# VisionTrim: Unified Vision Token Compression for Training-Free MLLM Acceleration

**Hanxun Yu**[1*], **Wentong Li**[2*], **Xuan Qu**[1*], **Song Wang**[1], **Junbo Chen**[3†], **Jianke Zhu**[1,4†]

[1]State Key Lab of CAD & CG, Zhejiang University    [2]NUAA    [3]Udeer.ai
[4]Shenzhen Loop Area Institute

[*]Equal contribution. [†]Corresponding authors.

## Abstract

Multimodal large language models (MLLMs) suffer from high computational costs due to excessive visual tokens, particularly in high-resolution and video-based scenarios. Existing token reduction methods typically focus on isolated pipeline components and often neglect textual alignment, leading to performance degradation. In this paper, we propose VisionTrim, a unified framework for training-free MLLM acceleration, integrating two effective plug-and-play modules: 1) the Dominant Vision Token Selection (DVTS) module, which preserves essential visual tokens via a global-local view, and 2) the Text-Guided Vision Complement (TGVC) module, which facilitates context-aware token merging guided by textual cues. Extensive experiments across diverse image and video multimodal benchmarks demonstrate the performance superiority of our VisionTrim, advancing practical MLLM deployment in real-world applications. The code is available at: https://github.com/hanxunyu/VisionTrim.

## 1 Introduction

With the recent advancements in large language models (LLMs) (Vicuna, 2023; Touvron et al., 2023; Bai et al., 2023a; Achiam et al., 2023), significant efforts (Bai et al., 2023b; Chen et al., 2024c; Reid et al., 2024) have been devoted to extending their impressive reasoning and interaction capabilities to vision-language tasks. Current multimodal large language models (MLLMs) typically integrate visual signals as sequential tokens, which are processed by an LLM to enable visual perception of the world.

Despite their promising performance, the extensive use of visual tokens, which dominate the input sequence of LLMs, substantially increases the computational complexity and cost associated with inference in MLLMs. This issue is particularly pronounced in high-resolution methods (Liu et al., 2024a;b; Chen et al., 2024c; Li et al., 2024b) and video-based models (Zhang et al., 2024b; Cheng et al., 2024; Shen et al., 2024), where the increased token length exacerbates computational overhead and severely restricts the practical deployment potential of VLMs (Jin et al., 2024b).

Recent studies (Wang et al., 2024a; 2025a; Ye et al., 2025; Zhong et al., 2024; Jin et al., 2025) have focused on accelerating the inference of MLLMs by reducing visual tokens while preserving essential information. For instance, FasterVLM (Zhang et al., 2024a) and VisionZip (Yang et al., 2025) perform global dominant visual token selection after vision encoding, whereas FastV (Chen et al., 2024a) and SparseVLM (Zhang et al., 2025b) prune tokens based on attention weights during LLM decoding. While these methods yield promising results, they tend to focus primarily on specific individual components of the MLLM framework, typically either the vision encoding or LLM decoding phases. Though the concurrent work VScan (Zhang et al., 2025a) adopts a two-stage pruning approach, it overlooks the essential role of the text query in aiding visual token selection during the vision encoding stage and directly uses the attention distribution between all visual tokens and the final instruction token for pruning during the LLM decoding stage, causing the potential loss of crucial text-related visual tokens. Furthermore, existing text-agnostic approaches like Pyramid-Drop (Xing et al., 2024) frequently overlook the necessity of aligning visual token selection with

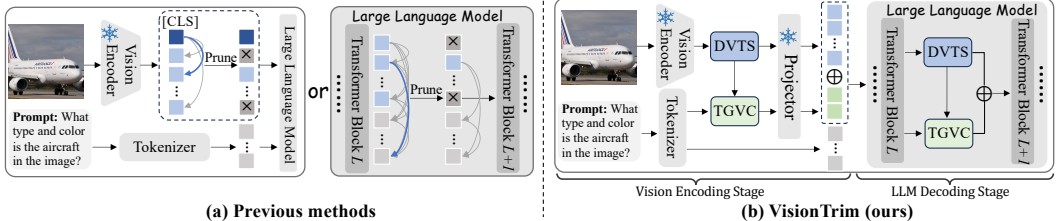

Figure 1: **Comparison of previous methods with VisionTrim.** (a) Previous methods focus solely on a specific part of the MLLM framework, typically the vision encoding or LLM decoding stages. (b) In contrast, VisionTrim optimizes the entire MLLM pipeline by introducing two plug-and-play modules, Dominant Vision Token Selection (DVTS) and Text-Guided Vision Complement (TGVC), to effectively reduce visual tokens in both the vision encoding and LLM decoding phases.

textual information. This oversight can result in the loss of textual context, which is essential for accurate LLM decoding, ultimately leading to a substantial degradation in performance.

To tackle these issues, we propose a unified vision token compression framework, named **VisionTrim**, for training-free acceleration of MLLMs. As illustrated in Figure 1, in contrast to previous methods that focus exclusively on visual token compression during either vision encoding or LLM decoding, our approach considers the entire forward propagation of the MLLM. We introduce two plug-and-play modules that effectively accelerate both the vision encoding and LLM decoding processes, which can be seamlessly inserted between any two layers of the vision encoder and the LLM. Specifically, our proposed method primarily consists of two key components: the Dominant Vision Token Selection (DVTS) and the Text-Guided Vision Complement (TGVC) modules.

Firstly, within the DVTS module, we consider both global semantics and local spatial continuity to filter visual tokens that convey essential visual information. Beyond utilizing [CLS] token's attention scores for global semantic importance, we develop the Local Token Affinity Measurement (LTAM) algorithm to simultaneously capture feature similarity and spatial proximity among visual tokens. This approach ensures that critical visual details are retained while reducing redundancy. Secondly, in the TGVC module, we leverage textual information to guide the clustering and merging of pruned visual tokens relevant to the input text instructions. These tokens are then employed to complement the dominant visual tokens from the DVTS module. By integrating textual context into the visual token reduction process, our method enhances the implicit alignment between visual and textual representations, thereby improving the overall efficiency and performance of the pruned MLLM. As shown in Figure 2, our approach consistently surpasses previous techniques across a range of reduction ratios, offering significant advantages in both efficiency and accuracy for various image- and video-based MLLMs. In summary, the contributions of our work are threefold:

- We introduce VisionTrim, a unified framework for vision token compression that enables training-free MLLM acceleration, optimizing the entire MLLM pipeline.
- We present two effective plug-and-play modules, DVTS and TGVC, designed to accelerate the forward processes of both the vision encoder and the LLM backbone, seamlessly integrable between any two layers.
- Extensive experiments conducted on a variety of multimodal benchmarks, spanning both standard and high-resolution, as well as image- and video-based MLLMs, clearly demonstrate the superiority of our VisionTrim over previous state-of-the-art counterparts.

## 2 RELATED WORK

### 2.1 MULTIMODAL LARGE LANGUAGE MODELS

Large Language Models (LLMs) (Vicuna, 2023; Touvron et al., 2023; Bai et al., 2023a; Team, 2023; Achiam et al., 2023) have garnered significant attention due to their powerful capabilities in natural language processing tasks such as text understanding, generation, and question answering. Nonetheless, the reliance on purely textual data limits their applicability, as human perception is inherently multimodal. This has spurred the development of Multimodal LLMs (MLLMs) (Liu

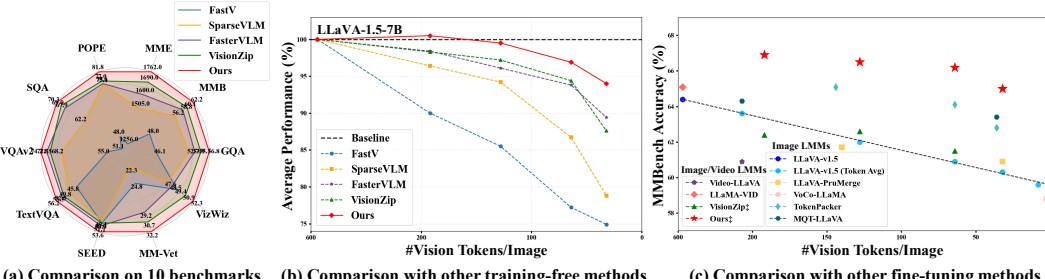

Figure 2: **Performance of VisionTrim.** (a) Comparison across 10 benchmarks using the standard LLaVA-1.5-7B (Liu et al., 2024a), with an **88.9%** reduction in visual tokens. (b) & (c) Performance *vs.* efficiency of various methods with a range of visual tokens, in both training-free and fine-tuning scenarios, respectively. The fine-tuning VisionTrim (Ours‡) demonstrates superior performance over previous image- and video-based MLLMs.

et al., 2023; Bai et al., 2023b; Chen et al., 2024c; Reid et al., 2024; Yu et al., 2025; Lin et al., 2024; Li et al., 2024a; Liu et al., 2024c; Wang et al., 2025b), which integrate LLMs with visual encoders to augment performance in multimodal tasks. The typical image- and video-based MLLMs (Liu et al., 2024a; Cheng et al., 2024; Lin et al., 2023) utilize an MLP to project visual information encoded by a Vision Transformer (ViT) (Dosovitskiy, 2020) into a space interpretable by LLMs, improving performance on visual-language tasks through visual instruction tuning. However, this paradigm requires a large number of visual tokens to represent visual information, particularly with high-resolution images and long-context video inputs, which further exacerbates the issue. The resulting increase in computational demands and inference times poses significant challenges, hindering the practical deployment of MLLMs in real-world applications.

## 2.2 VISION TOKEN COMPRESSION FOR MLLMS

The quadratic complexity inherent in Transformer networks (Vaswani et al., 2017), which scales with the sequence length of input tokens in MLLMs, remains a widely acknowledged challenge. To address this issue, several methods (Li et al., 2023a; Bai et al., 2023b; Cha et al., 2024; Li et al., 2024b; Yao et al., 2024; Hu et al., 2024; Chu et al., 2023; 2024) explore efficient visual projectors that enable compact visual representations using fewer visual tokens before feeding them into the LLM. While these approaches have demonstrated promising performance, they often necessitate architectural modifications and extensive training. Alternatively, recent works (Shang et al., 2024; Chen et al., 2024a; Zhang et al., 2024a; 2025b; Yang et al., 2025) aim to reduce visual tokens in a training-free manner and mainly focus on either the vision encoding or LLM decoding stages. LLaVA-PruMerge (Shang et al., 2024) utilizes class-spatial similarity for pruning, and Faster-VLM (Zhang et al., 2024a) evaluates token importance via attention scores between the `[CLS]` token and image tokens, both operating before sending the vision tokens to the LLM. FastV (Chen et al., 2024a) and SparseVLM (Zhang et al., 2025b) prune redundant tokens at a specific layer of LLM based on attention scores solely during the LLM decoding stage. Additionally, most existing approaches overlook the alignment between visual token selection and textual information. While CrossGET (Shi et al., 2024) and Turbo (Ju et al., 2024) directly leverage text-visual attention to aid token selection, they place excessive emphasis on text tokens, which can lead to hallucinations and disrupt multi-round interactions. In contrast, our approach considers the entire MLLM pipeline and simultaneously integrates both global semantic significance and local spatial continuity to preserve visual integrity. Furthermore, we introduce a text-guided visual complement mechanism to ensure alignment with textual instructions, offering a more comprehensive and effective solution to the challenge of vision token compression.

## 3 METHODOLOGY

As illustrated in Figure 3, our approach comprehensively considers the entire pipeline of MLLM, comprising two key components that simultaneously accelerate the vision encoder and LLM forward processes. The first component, Dominant Vision Token Selection (DVTS) module, meticulously fil-

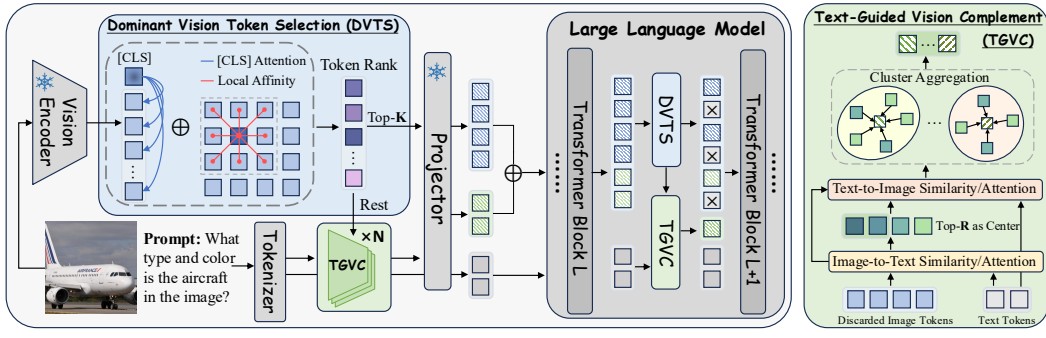

**(a) Overall training-free framework**       **(b) Structure of TGVC module**

Figure 3: (a) Overview of VisionTrim featuring the detailed DVTS module, and (b) the structure of the TGVC module. Both DVTS and TGVC modules can be generally utilized in both the vision encoding stage and the LLM decoding stage.

ters tokens to preserve vital visual information, focusing particularly on their significance for global semantics and local spatial continuity. The second component, Text-Guided Vision Complement (TGVC) module, leverages textual context to guide the clustering and merging of discarded visual tokens relevant to the input text instructions. This process complements the dominant visual tokens by integrating critical visual details. Both DVTS and TGVC are designed as plug-and-play modules that can be seamlessly integrated between any two layers of either the vision encoder or the LLM.

## 3.1 DOMINANT VISION TOKEN SELECTION (DVTS)

To preserve visual integrity during visual token compression, we introduce a novel scoring mechanism for selecting dominant vision tokens. This mechanism thoroughly incorporates both global semantic significance and local spatial continuity. Initially, we utilize [CLS] token's attention scores relative to other visual tokens to assess global semantic importance. Then, we develop the Local Token Affinity Measurement (LTAM) algorithm, which employs a dual-kernel method to capture feature similarity and spatial proximity, thereby ensuring local spatial continuity. These complementary metrics are subsequently integrated using an adaptive variance-based weighting scheme to prioritize the reliable visual tokens.

**Global Semantic Importance.** Motivated by previous methods (Yang et al., 2025; Zhang et al., 2024a), the [CLS] token's attention distribution across all image tokens serves as a natural measure of global semantic significance. We extract the attention weights from the penultimate layer of the CLIP-based vision encoder and leverage the attention patterns from the [CLS] token. The self-attention computation for the [CLS] token is expressed as follows:

$$\mathbf{Q}_{[CLS]} = \mathbf{W_Q} X^{L-1}_{[CLS]}, \quad \mathbf{K}_i = \mathbf{W_K} X^{L-1}_i, \tag{1}$$

$$A^{L-1}_{[CLS],i} = \mathrm{softmax}(\mathbf{Q}_{[CLS]}\mathbf{K}_i^T/\sqrt{d_k}), \ \ i \in [1, N]. \tag{2}$$

Here, $X^{L-1}_{[CLS]}$ and $X^{L-1}_i$ denote the hidden states of the [CLS] token and the $i$-th visual token at the $(L-1)$-th layer, respectively. $\mathbf{W_Q}$ and $\mathbf{W_K}$ are learnable projection matrices, and $d_k$ is the dimension of key vector. $N$ represents the total number of visual tokens. The global importance score $S^g_i$ for the $i$-th visual token is the average attention score across all $H$ heads:

$$S^g_i = \frac{1}{H} \sum_{h=1}^{H} A^{L-1}_{[CLS],i,h}, \quad i \in [1, N]. \tag{3}$$

This formulation effectively assesses each visual token's contribution to the global semantic representation of the image based on the [CLS] token's attention mechanism. The computed global scores $\{S^g_i\}^N_{i=1}$ are then normalized to yield a probability distribution over all visual tokens, *i.e.* $\hat{S}^g_i = \exp(S^g_i)/\sum_{j=1}^{N} \exp(S^g_j)$.

**Local Spatial Continuity.** Inspired by (Ru et al., 2022; Li et al., 2023b), we introduce the Local Token Affinity Measurement (LTAM) algorithm to effectively capture the local spatial continuity of

visual tokens. LTAM utilizes a dual-kernel affinity mechanism to simultaneously account for feature similarity and positional proximity. For the $i$-th token at position $(x, y)$, its local importance $S_i^l$ is determined by computing the affinity with neighboring tokens within a local kernel $\mathcal{N}(x, y)$ of size $k \times k$. For tokens positioned at $(x, y)$ and $(u, v)$, the affinity kernel $\kappa^*$ is defined as a weighted combination of a feature-based term $\kappa_{feat}$ and a position-based term $\kappa_{pos}$:

$$\kappa_{feat}^{xy,uv} = -\left(\frac{\|F_{xy} - F_{uv}\|}{w_1 \sigma_f}\right)^2, \kappa_{pos}^{xy,uv} = -\left(\frac{\|P_{xy} - P_{uv}\|}{w_2 \sigma_p}\right)^2, \tag{4}$$

$$\kappa^{*xy,uv} = \kappa_{feat}^{xy,hw} + w_3 \kappa_{pos}^{xy,hw}, \tag{5}$$

where $F_{xy} \in \mathbb{R}^d$ and $P_{xy} \in \mathbb{R}^2$ denote the feature vector and spatial coordinates of the token at $(x, y)$, respectively. $\sigma_f$ and $\sigma_p$ represent the standard deviations of the feature and positional differences. The pair $(h, w)$ is sampled from the neighborhood set $\mathcal{N}(x, y)$, and $w_1$, $w_2$, and $w_3$ are balancing parameters. The local importance $S_i^l$ of the $i$-th token at $(x, y)$ is then computed by averaging the affinity scores $\kappa^*$ over all neighboring tokens and converting to a probability distribution.

**Adaptive Variance-based Weighting.** To integrate global and local importance scores, we present an adaptive variance-based weighting mechanism:

$$S_i = \alpha \hat{S}_i^g + (1 - \alpha)S_i^l, \quad \text{where} \quad \alpha = \sigma_l^2 / (\sigma_g^2 + \sigma_l^2). \tag{6}$$

$\sigma_g^2$ and $\sigma_l^2$ denote the variances of the global and local importance scores, respectively. This adaptive weighting scheme automatically prioritizes more reliable signals based on their consistency, ensuring robust token selection. The final importance scores, $\{S_i\}_{i=1}^N$, are used to select the top-$K$ informative tokens $\mathbf{V}_{dom} \in \mathbb{R}^{K \times d}$ from the complete set $\mathbf{V} \in \mathbb{R}^{N \times d}$. This selection process ensures the preservation of both semantic relevance and spatial continuity.

## 3.2 Text-Guided Vision Complement (TGVC)

Selected dominant tokens, while capturing primary visual information, may not fully reflect their relevance to the input instructions, potentially leading to misalignment with the textual information and loss of crucial visual elements. To address this issue, we introduce the Text-Guided Vision Complement (TGVC) module, which utilizes text instructions to complement the selected dominant vision tokens. By leveraging CLIP's text encoder, we calculate the similarity between the remaining visual tokens and text tokens, identifying the top-$R$ tokens as clustering centers. These centers then direct the allocation of remaining visual tokens to $R$ clusters. Each cluster is merged to yield the final $R$ visual tokens most relevant to the text, which we term the vision complement tokens.

**Clustering Centers.** Given the remaining visual tokens $\mathbf{V}_r \in \mathbb{R}^{(N-K) \times d}$ after dominant token selection, we begin by calculating their similarity $S_{t2v} \in \mathbb{R}^{L \times (N-K)}$ with the text features $T \in \mathbb{R}^{L \times d}$ to identify potential clustering centers:

$$S_{t2v} = \text{softmax}(T \mathbf{V}_r^T / \sqrt{d}). \tag{7}$$

Next, token-level importance scores $s \in \mathbb{R}^{N-K}$ are obtained by averaging the similarity scores across all text tokens, expressed as $s = \frac{1}{L} \sum_{i=1}^L S_{t2v_i}$. The top-$R$ tokens are then selected as clustering centers, denoted as $C = \{c_1, ..., c_R\}$.

**Token Assignment.** For each remaining token $v_i \in \mathbf{V}_r \setminus C$, we compute its assignment score for each clustering center using text-guided similarity. Specifically, for a center $c_j$, the similarity scores are calculated as follows:

$$S_{v2t}^i = \text{softmax}(v_i T^T / \sqrt{d}), \quad S_{t2c}^j = \text{softmax}(T c_j^T / \sqrt{d}). \tag{8}$$

The assignment score $a_{ij}$ is then determined by $a_{ij} = S_{v2t}^i S_{t2c}^j$. Each token is assigned to the clustering center with the highest similarity score:

$$\text{cluster}(v_i) = \arg\max_j a_{ij}. \tag{9}$$

**Cluster Aggregation.** For each cluster centered at $c_j$, we aggregate the assigned tokens through weighted averaging based on their text-guided similarities:

$$v_j^{\text{com}} = c_j + \sum_{v_i \in \text{cluster}(j)} \frac{a_{ij}}{\sum_{v_k \in \text{cluster}(j)} a_{kj}} v_i. \tag{10}$$

Table 1: Comparison with other methods on LLaVA-1.5-7B. The vanilla visual token count is 576. The best results in each setting are **bolded**, and the second-best are underlined.

| Method | GQA | MMB | MME | POPE | SQA | VQA$^{\text{V2}}$ | VQA$^{\text{Text}}$ | SEED | MMVet | VizWiz | Avg. |
|---|---|---|---|---|---|---|---|---|---|---|---|
| | | | *Upper Bound, 576 Tokens* | | | (**100%**) | | | | | |
| Vanilla | 61.9 | 64.7 | 1862 | 85.9 | 69.5 | 78.5 | 58.2 | 58.6 | 31.1 | 50.1 | 100.0% |
| | | | *Retain Averaged 192 Tokens* | | | (↓66.7%) | | | | | |
| SparseVLM | 57.6 | 62.5 | 1721 | 83.6 | 69.1 | 75.6 | 56.1 | 55.8 | 30.2 | 50.0 | 96.4% |
| VisionZip | 59.3 | 63.0 | 1783 | 85.3 | 68.9 | 76.8 | 57.3 | 56.4 | 31.7 | 50.5 | 98.3% |
| PDrop | 57.3 | 63.3 | 1797 | 84.8 | 69.2 | 76.4 | 56.5 | 57.2 | 30.8 | 50.0 | 97.6% |
| VScan | 60.6 | 63.9 | **1806** | 86.2 | 68.6 | 77.8 | 57.7 | – | – | 50.4 | 98.9% |
| **Ours** | **61.0** | **64.4** | 1798 | **86.8** | **70.8** | **78.4** | **58.4** | **58.3** | **33.2** | **51.3** | **100.6%** |
| | | | *Retain Averaged 128 Tokens* | | | (↓77.8%) | | | | | |
| SparseVLM | 56.0 | 60.0 | 1696 | 80.5 | 67.1 | 73.8 | 54.9 | 53.4 | 30.0 | 51.0 | 94.2% |
| VisionZip | 57.6 | 62.0 | 1762 | 83.2 | 68.9 | 75.6 | 56.8 | 54.9 | 32.6 | 50.0 | 97.2% |
| PDrop | 57.1 | 61.6 | 1761 | 82.6 | 68.4 | 76.0 | 56.6 | 56.2 | 31.0 | 49.6 | 96.5% |
| VScan | 59.8 | 63.0 | **1792** | 86.1 | 68.9 | 77.1 | 57.3 | – | – | **51.7** | 98.6% |
| **Ours** | **60.3** | **64.0** | 1788 | **86.6** | **69.7** | **78.2** | **58.2** | **57.5** | **32.7** | 51.4 | **99.9%** |
| | | | *Retain Averaged 64 Tokens* | | | (↓88.9%) | | | | | |
| SparseVLM | 52.7 | 56.2 | 1505 | 75.1 | 62.2 | 68.2 | 51.8 | 51.1 | 23.3 | 49.6 | 86.7% |
| VisionZip | 55.1 | 60.1 | 1690 | 77.0 | 69.0 | 72.4 | 55.5 | 52.2 | 31.7 | 51.9 | 94.4% |
| PDrop | 47.5 | 58.8 | 1561 | 76.2 | 69.0 | 73.3 | 50.6 | 53.0 | 30.5 | 50.2 | 90.8% |
| VScan | 58.3 | 62.1 | 1698 | 85.0 | 69.1 | 75.4 | 55.6 | – | – | 51.8 | 96.8% |
| **Ours** | **58.8** | **63.0** | **1780** | **86.2** | **71.0** | **76.8** | **56.8** | **54.8** | **32.2** | **52.3** | **98.8%** |

This process is repeated for $T$ iterations to refine the clusters. The final vision complement tokens $\mathbf{V}_{com} = \{v_1^{com}, v_2^{com}, ..., v_R^{com}\}$ are then concatenated with the dominant tokens to form the complete visual representation:

$$\mathbf{V}_{final} = [\mathbf{V}_{dom}; \mathbf{V}_{com}] \in \mathbb{R}^{(K+R) \times d}. \qquad (11)$$

This text-guided complement mechanism ensures that visual tokens effectively capture key visual details of the image while remaining aligned with the textual instruction.

## 3.3 MULTI-STAGE PRUNING STRATEGY

Our Dominant Vision Token Selection (DVTS) and Text-Guided Vision Complement (TGVC) modules provide a versatile approach to token reduction that can be effectively applied at two stages of the MLLM pipeline.

1) **Vision Encoding Stage**: Before LLM processing, DVTS and TGVC can reduce the initial visual token sequence $\mathbf{V} = \{\mathbf{v}_1, \mathbf{v}_2, ..., \mathbf{v}_N\}$ to a more compact representation $\mathbf{V}' = \{\mathbf{v}'_1, \mathbf{v}'_2, ..., \mathbf{v}'_{K+R}\}$, where $K + R < N$.

2) **LLM Decoding Stage**: DVTS and TGVC can be integrated between any two transformer layers during LLM decoding, enabling dynamic token pruning while preserving cross-modal alignment. Specifically, instead of using the [CLS] token, we leverage the attention distribution of the first generated token as a natural measure of the global semantic significance over all image tokens. At layer $l$, global semantic scores $\mathbf{S}^g$ for DVTS and cross-modal attention scores $\mathbf{A}$ between visual and textual tokens for TGVC are computed as follows:

$$\mathbf{S}^g = \text{softmax}\left(\frac{\mathbf{H}_{gen}^l \mathbf{H}_v^l}{\sqrt{D}}\right) \in \mathbb{R}^{1 \times N_v}, \quad \mathbf{A} = \text{softmax}\left(\frac{\mathbf{H}_v^l \mathbf{H}_t^l}{\sqrt{D}}\right) \in \mathbb{R}^{N_v \times N_t}, \quad \alpha_i = \frac{1}{N_t}\sum_{j=1}^{N_t} \mathbf{A}_{i,j}, \quad (12)$$

where $\mathbf{H}_{gen}^l \in \mathbb{R}^{1 \times D}$, $\mathbf{H}_v^l \in \mathbb{R}^{N_v \times D}$ and $\mathbf{H}_t^l \in \mathbb{R}^{N_t \times D}$ represent the first generated token, visual tokens, and textual tokens at layer $l$, respectively. $\alpha_i$ denotes the average cross-modal attention score for the $i$-th visual token. Using these scores along with the local spatial affinity scores $\mathbf{S}^l$ from the LTAM mechanism, we first select the top-$K$ tokens $\mathbf{V}_{dom}$ in DVTS and then perform top-$R$ token complement $\mathbf{V}_{com}$ in TGVC. Finally, we obtain $\mathbf{V}_{final} = [\mathbf{V}_{dom}; \mathbf{V}_{com}]$. The multi-stage application of our proposed DVTS and TGVC modules refines the visual representation, while ensuring both computational efficiency and effective cross-modal alignment.

Table 2: Performance comparisons across various token counts on LLaVA-NeXT-7B.

| Method | GQA | MMB | MME | SQA | VQA$^T$ | POPE | Avg. |
|---|---|---|---|---|---|---|---|
| *Upper Bound, 2880 Tokens* | | | (**100%**) | | | | |
| Vanilla | 64.2 | 67.9 | 1842 | 70.2 | 61.3 | 86.3 | 100% |
| *Retain 640 Tokens* | | | (↓ 77.8%) | | | | |
| SparseVLM | 60.3 | 65.7 | 1772 | 67.7 | 57.8 | 85.2 | 96.1% |
| | 93.9% | 96.8% | 96.2% | 96.4% | 94.3% | 98.7% | |
| VisionZip | 61.3 | 66.3 | 1787 | 68.1 | 60.2 | 87.7 | 97.8% |
| | 95.5% | 97.6% | 97.0% | 97.0% | 98.2% | 101.6% | |
| PDrop | 62.9 | 66.5 | 1733 | 69.4 | 58.3 | 86.4 | 97.4% |
| | 98.0% | 97.9% | 94.1% | 98.9% | 95.1% | 100.1% | |
| **Ours** | **63.2** | **67.2** | **1825** | **70.7** | **61.0** | **88.5** | **99.9%** |
| | 98.4% | 99.0% | 100.7% | 100.7% | 99.5% | 102.5% | ↑ (2.1%) |
| *Retain 320 Tokens* | | | (↓ 88.9%) | | | | |
| SparseVLM | 57.7 | 64.3 | 1694 | 67.0 | 55.9 | 78.6 | 92.4% |
| | 89.9% | 94.7% | 92.0% | 95.4% | 91.2% | 91.1% | |
| VisionZip | 59.3 | 63.1 | 1702 | 67.3 | 58.9 | 82.1 | 94.1% |
| | 92.4% | 92.9% | 92.4% | 95.9% | 96.1% | 95.1% | |
| PDrop | 58.5 | 63.2 | 1667 | 66.8 | 58.3 | 81.9 | 93.3% |
| | 91.1% | 93.1% | 90.5% | 95.2% | 94.9% | 94.9% | |
| **Ours** | **61.7** | **64.8** | **1795** | **69.6** | **59.6** | **83.6** | **97.0%** |
| | 96.1% | 95.4% | 97.4% | 99.1% | 97.2% | 96.9% | ↑ (2.9%) |
| *Retain 160 Tokens* | | | (↓ 94.4%) | | | | |
| SparseVLM | 51.2 | 63.1 | 1542 | 67.5 | 46.4 | 77.3 | 86.3% |
| | 79.8% | 92.9% | 83.7% | 96.2% | 75.7% | 89.6% | |
| VisionZip | 55.5 | 60.1 | 1630 | 68.3 | 56.2 | 79.4 | 90.7% |
| | 86.4% | 88.5% | 88.5% | 97.3% | 91.7% | 92.0% | |
| PDrop | 56.1 | 60.3 | 1545 | 67.4 | 54.7 | 78.0 | 89.3% |
| | 87.4% | 88.8% | 83.9% | 96.0% | 89.2% | 90.4% | |
| **Ours** | **57.2** | **63.3** | **1702** | **70.2** | **58.3** | **81.1** | **94.0%** |
| | 89.1% | 93.2% | 92.4% | 100.0% | 95.1% | 94.0% | ↑ (3.3%) |

Table 3: Comparison with previous state-of-the-art methods on Video-LLaVA-7B.

| Method | TGIF | | MSVD | | MSRVTT | | ActivityNet | | Avg. | |
|---|---|---|---|---|---|---|---|---|---|---|
| | Acc | Score | Acc | Score | Acc | Score | Acc | Score | Acc | Score |
| Vanilla | 47.1 | 3.35 | 69.8 | 3.92 | 56.7 | 3.48 | 43.1 | 3.35 | 100.0% | +0.00 |
| SparseVLM | 44.7 | 3.29 | 68.2 | 3.90 | 31.0 | 2.68 | 42.6 | 3.32 | 86.5% | -0.23 |
| | 94.9% | -0.06 | 97.7% | -0.02 | 54.7% | -0.80 | 98.8% | -0.03 | | |
| VisionZip | 42.0 | 3.16 | 63.5 | 3.58 | 49.6 | 3.34 | 42.0 | 3.21 | 91.3% | -0.20 |
| | 89.2% | -0.19 | 91.0% | -0.34 | 87.5% | -0.14 | 97.4% | -0.14 | | |
| **Ours** | **45.2** | **3.32** | **68.6** | **3.93** | **54.9** | **3.42** | **43.5** | **3.31** | **98.0%** | **-0.03** |
| | 96.0% | -0.03 | 98.3% | 0.01 | 96.8% | -0.06 | 100.9% | -0.04 | ↑ (6.7%) | ↑ (0.17) |

Table 4: Ablation of pruning at vision encoding and LLM decoding stages.

| Stages | #Tokens | GQA | MMB | POPE | VQA$^{V2}$ | KV Cache (MB) |
|---|---|---|---|---|---|---|
| LLaVA-1.5-7B | 576 | 61.9 | 64.7 | 85.9 | 78.5 | 303.6 |
| **Only in ViT** | | | | | | |
| *w/* DVTS | 64 | 52.8 | 56.9 | 76.1 | 68.6 | **25.4** (↓ 91.6%) |
| *w/* DVTS+TGVC | | 55.6 | 60.2 | 80.2 | 72.2 | |
| **Only in LLM** | | | | | | |
| *w/* DVTS | 64 | 52.0 | 57.4 | 75.8 | 70.2 | 43.5 (↓ 85.7%) |
| *w/* DVTS+TGVC | | 54.7 | 61.1 | 79.2 | 73.6 | |
| **Both in ViT and LLM** | 64 | **58.8** | **63.0** | **86.2** | **76.8** | 30.2 (↓ 90.1%) |

Table 5: Ablation study of various ensemble strategies in the DVTS module.

| Ensemble Strategy | GQA | MMB | MME | POPE | SQA | VQA$^{Text}$ |
|---|---|---|---|---|---|---|
| Only [CLS] token | 52.8 | 55.3 | 1536 | 74.2 | 67.9 | 51.2 |
| Element-wise Maximum | 53.4 | 58.7 | 1702 | 77.6 | 70.0 | 52.4 |
| Geometric Mean | 55.2 | 56.5 | 1631 | 80.2 | **71.1** | 54.3 |
| Adaptive Weighting | **58.8** | **63.0** | **1780** | **86.2** | 71.0 | **56.8** |

# 4 EXPERIMENT

## 4.1 EXPERIMENTAL SETTINGS

**Datasets and Benchmarks.** We conduct a comprehensive evaluation across 10 widely-used image-based benchmarks to assess the multimodal understanding and reasoning capabilities of our proposed approach. These benchmarks include common visual question answering tasks, like GQA (Hudson & Manning, 2019), VQA$^{V2}$ (Goyal et al., 2017) and VizWiz (Gurari et al., 2018), as well as other multimodal benchmarks such as POPE (Li et al., 2023c), MMBench (Liu et al., 2025), MME (Fu et al., 2023) and MM-Vet (Yu et al., 2023). Additionally, we experiment with 4 widely used video-based multimodal understanding tasks: TGIF-QA (Jang et al., 2017), MSVD-QA (Xu et al., 2017), MSRVTT-QA (Xu et al., 2017), and ActivityNet-QA (Yu et al., 2019).

**Implementaion Details.** We apply our approach to various open-source MLLMs, including the classic LLaVA-1.5 (Liu et al., 2024a) model for normal-resolution images, LLaVA-NeXT (Liu et al., 2024b) for high-resolution images, Video-LLaVA (Lin et al., 2023) for video-based tasks, and Qwen2-VL (Wang et al., 2024b) and Qwen2.5-VL (Bai et al., 2025) for broader validation. To ensure a fair comparison, we adopt the default settings and evaluation metrics as reported in their respective papers. We compare our approach with SparseVLM (Zhang et al., 2025b), VisionZip (Yang et al., 2025), PyramidDrop (Xing et al., 2024), and VScan (Zhang et al., 2025a). Following the same spirit, we design different algorithms for multiple stages of the MLLM pipeline.

## 4.2 MAIN RESULTS

**Normal Resolution.** As shown in Table 1, we first evaluate our approach on LLaVA-1.5-7B (Liu et al., 2024a) under the normal-resolution setting. VisionTrim consistently surpasses previous methods across all token configurations (192, 128, and 64). In benchmarks, such as POPE, SQA, and TextVQA, VisionTrim not only maintains its performance without degradation but also achieves improvements, highlighting the severe redundancy present in visual tokens fed to the LLM.

**High Resolution.** Our approach minimizes token count while maintaining performance on LLaVA-NeXT-7B (Liu et al., 2024b) with high-resolution inputs. As demonstrated in Table 2, VisionTrim retains 99.9% of the original performance using only 22.2% visual tokens. With nearly 95% token

Table 6: Experiment results of deploying VisionTrim on Qwen2-VL-7B and Qwen2.5-VL-7B over several benchmarks. For both models, approximately 1/3 of the original input tokens are used.

| Method | MMB | MMStar | MME | VQA$^T$ | POPE |
|---|---|---|---|---|---|
| Qwen2-VL | 80.7 | 60.7 | 2322 | 84.3 | 86.4 |
| FastV | 77.8 | 57.3 | 1859 | 77.6 | 82.0 |
| SparseVLM | 79.0 | 57.9 | 2019 | 79.2 | 84.2 |
| PDrop | 80.6 | 58.6 | 2053 | 80.2 | 82.5 |
| **Ours** | **82.8** | **60.6** | **2310** | **83.5** | **86.3** |

| Method | MMB | MMB$^{CN}$ | MMMU | SEED | RefCOCO |
|---|---|---|---|---|---|
| Qwen2.5-VL | 83.5 | 83.4 | 38.3 | 70.4 | 89.5 |
| FastV | 77.5 | 76.3 | 34.6 | 64.8 | 73.6 |
| SparseVLM | 79.6 | 80.3 | 36.0 | 67.5 | 78.4 |
| PDrop | 78.2 | 77.0 | 35.8 | 67.9 | 77.5 |
| **Ours** | **83.2** | **81.8** | **37.9** | **70.2** | **86.8** |

Table 7: Ablation study on TGVC module. This experiment, conducted before inputting data into the LLM, evaluates the effectiveness of the TGVC in reducing noise within text-visual attention during the LLM's forward pass.

| Benchmark | Number of Tokens | | | | Avg. |
|---|---|---|---|---|---|
| | 192 | 128 | 64 | 32 | |
| POPE | 83.0 | 81.4 | 76.1 | 72.9 | 78.4 |
| w/ TGVC | 86.1 (↑ 3.1) | 84.7 (↑ 3.3) | 80.2 (↑ 4.1) | 77.3 (↑ 4.4) | 82.1 (↑ 3.7) |
| MMBench | 61.1 | 60.5 | 56.9 | 54.9 | 58.4 |
| w/ TGVC | 63.4 (↑ 2.3) | 62.9 (↑ 2.4) | 60.2 (↑ 3.3) | 59.1 (↑ 4.2) | 61.4 (↑ 3.0) |
| Text-VQA | 55.3 | 54.4 | 52.6 | 50.2 | 53.1 |
| w/ TGVC | 57.8 (↑ 2.5) | 57.2 (↑ 2.8) | 56.0 (↑ 3.4) | 54.2 (↑ 4.0) | 56.3 (↑ 3.2) |

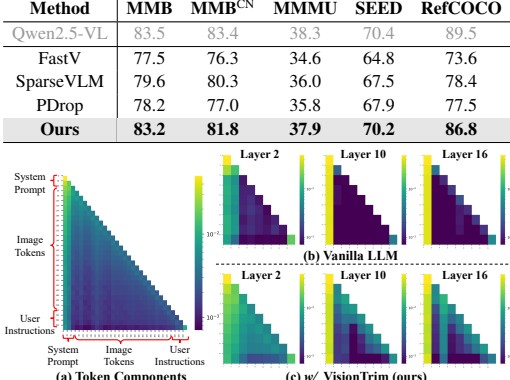

Figure 4: Comparison of attention maps during LLM forward processing, with and without our proposed VisionTrim.

reduction, it achieves 94.0% performance without training, surpassing the previous state-of-the-art method, VisionZip (Yang et al., 2025), by 3.3%. These results validate the superior efficacy of VisionTrim for high-resolution inputs.

**Video.** To assess the generalization of our approach on video-based scenarios, we apply it to Video-LLaVA-7B (Lin et al., 2023), which processes 8 frames from a video and generates 2048 visual tokens. Following SparseVLM (Zhang et al., 2025b), we reduce the visual tokens to 136. As shown in Table 3, VisionTrim achieves 98.0% of the original performance with a 93.4% pruning ratio, outperforming all other methods across four benchmarks. Furthermore, VisionTrim consistently exceeds 96.0% in performance, demonstrating its effectiveness and robustness. Our method excels even with high pruning ratios, effectively balancing inference speed and accuracy in video tasks.

**Broader Validation.** To further evaluate the effectiveness of VisionTrim, we deploy it to the state-of-the-art open-source MLLMs, Qwen2-VL-7B (Wang et al., 2024b) and Qwen2.5-VL-7B (Bai et al., 2025), using approximately 1/3 of the original input tokens. As shown in Table 6, VisionTrim exhibits competitive performance with only about a 0.1% performance loss across several cases, and even occasionally outperforms the baseline MLLM. Notably, VisionTrim exceeds the vanilla Qwen2-VL (Wang et al., 2024b) by 2.1% on the MMBench dataset, confirming its effectiveness in reducing visual redundancy. *Please refer to the Appendix for more experiments on other tasks to further assess VisionTrim's generalization capabilities.*

### 4.3 ABLATION STUDY

**Component-wise Analysis.** We conduct a thorough ablation study to evaluate our approach for both vision encoding and LLM decoding stages, as presented in Table 4. We reduce image tokens to 64 for an 88.9% reduction (the same below). Initially, applying DVTS and TGVC modules solely in the vision encoder improves multimodal processing and reduces KV cache memory by 91.6%. For a fair comparison, we also implement the DVTS and TGVC modules only in LLM's decoding stage, yielding performance gains of 5.4% and 4.9% over SparseVLM (Zhang et al., 2025b) on VQA$^{V2}$ and MMBech datasets, respectively. When applied to both vision encoding and LLM decoding stages, VisionTrim outperforms approaches that target only specific stages and achieves higher performance with a 90.1% reduction in memory usage, significantly surpassing existing state-of-the-art methods. Moreover, Figure 4 shows attention maps with and without VisionTrim, highlighting that the vanilla LLM exhibits high redundancy and suboptimal cross-modal alignment. In contrast, VisionTrim improves cross-modal alignment and reduces visual redundancy without performance compromise. *Please refer to the Appendix for more ablation studies and visualization results.*

**Ensemble Strategy in DVTS.** In the DVTS module, we employ various ensemble strategies to combine global semantic information from the [CLS] token attention and local spatial affinity captured by the LTAM algorithm, as shown in Table 5. Specifically, we explore three ensemble methods: element-wise maximum, geometric mean, and adaptive variance-based weighting. Compared to the

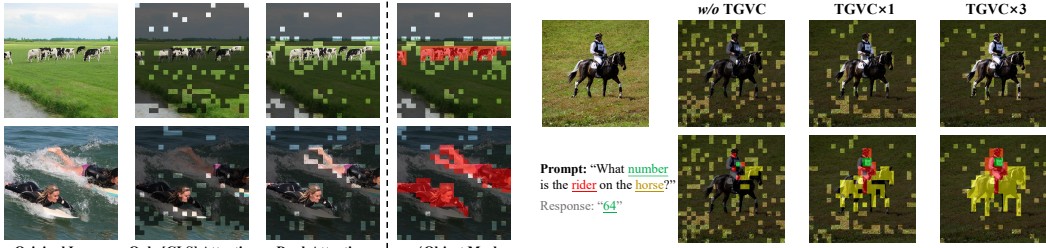

**Figure 5:** Visualization of retained visual patches with and without the dual-attention mechanism in the DVTS module. Black-masked areas indicate discarded visual tokens.

**Figure 6:** Visualization of retained visual patches with and without TGVC module. We show the correspondence between the salient visual regions and text in different colors.

baseline that solely uses [CLS] token attention, incorporating both global semantic and local spatial continuity enhances performance significantly. Furthermore, as depicted in Figure 5, relying exclusively on [CLS] token results in the loss of crucial semantic information, while considering local spatial continuity helps retain better visual token coverage. Consequently, our proposed DVTS with a dual-attention filtering mechanism offers a more holistic approach to attention integration.

**Visual Token Complement of TGVC.** As shown in Table 7, incorporating the TGVC module significantly boosts performance across three multimodal tasks: POPE, MMBench, and Text-VQA. Notably, as the compression ratio increases and token count decreases, the TGVC module's impact becomes more pronounced, resulting in performance gains exceeding 4%. Additionally, Figure 6 illustrates that the TGVC module retains essential visual tokens related to textual instructions, ensuring critical visual information is not pruned away. It can also be applied multiple times for enhanced visual completion, allowing textual tokens to better align with the corresponding visual information in the subsequent LLM decoding stage.

**The Usage of Textual Prompts.** In VisionTrim, token compression has two stages: dominant visual tokens are selected via a dual-attention mechanism, and discarded tokens are utilized with text-guided cues. (1) Without textual prompts, VisionTrim runs in a text-agnostic mode using DVTS, which relies on global [CLS] attention and local affinity; as shown in Table 10 in Appendix D.3, this maintains most accuracy while improving efficiency. (2) When prompts are irrelevant or misleading, text–image similarities become uniformly low, making textual initialization effectively random. Due to the semantic redundancy of visual tokens, the system behaves like unsupervised visual clustering (similar to ToMe (Bolya et al., 2022)), naturally grouping meaningful tokens. Since VisionTrim merges rather than prunes tokens, essential semantics are preserved even under poor textual cues.

## 4.4 EFFICIENCY ANALYSIS

We evaluate the efficiency of our method by measuring CUDA time, FLOPs, and storage memory, and compare it with vanilla LLaVA-NeXT-7B (Liu et al., 2024b) and other techniques, as shown in Table 8. At an 88.9% reduction ratio, our method reduces CUDA time by

Table 8: Efficiency analysis of our method on LLaVA-NeXT-7B. The detailed metric includes latency (CUDA time), computation (FLOPs), and storage (cache memory).

| Methods | #Tokens | SQA (%) ↑ | CUDA Time (Min & Sec) ↓ | △ | FLOPs (T) ↓ | △ | KV Cache (MB) ↓ | △ |
|---|---|---|---|---|---|---|---|---|
| Vanilla | 2880 | 70.2 | 26:34 | – | 9.6 | – | 1512.1 | – |
| SparseVLM | 320 | 67.0 | 18:26 | 30.6% | 1.5 | 84.4% | 168.0 | 88.9% |
| VisionZip | 320 | 67.3 | 17:53 | 32.7% | 1.6 | 83.3% | 180.4 | 88.1% |
| **Ours** | **320** | **69.6** | **10:16** | **61.4%** | **0.8** | **91.7%** | **101.8** | **93.3%** |

61.4%, FLOPs by 91.7%, and storage memory by 93.3%, while maintaining 99.1% accuracy on SQA. Notably, when retaining the same token count, our method is 44.3% faster in inference time compared to SparseVLM (Zhang et al., 2025b) and requires 50.0% less computational budget than VisionZip (Yang et al., 2025), while also minimizing KV cache memory usage. These results demonstrate the high efficiency of our approach. Additionally, as shown in Table 9, we provide further efficiency analyses on the POPE benchmark, including both the overall end-to-end latency and prefill time, demonstrating that VisionTrim is highly effective at accelerating MLLM inference.

Table 9: Additional efficiency results on the POPE benchmark. We further report the end-to-end latency, prefill time, and the corresponding accuracy on a single NVIDIA A100 GPU.

| Methods | #Tokens | Total Inference Time ↓ | Prefill Time ↓ | FLOPs ↓ | Accuracy ↑ |
|---|---|---|---|---|---|
| LLaVA-1.5-7B | 576 | 1303 s (1.00×) | 494 s (1.00×) | 3.8 T | 85.9 |
| + SparseVLM | 64 | 1068 s (1.22×) | 377 s (1.31×) | 1.3 T | 75.1 |
| **+ Ours** | 64 | **685 s (1.90×)** | **235 s (2.10×)** | **0.8 T** | **86.2** |
| LLaVA-NeXT-7B | 2880 | 2284 s (1.00×) | 1062 s (1.00×) | 12.6 T | 86.3 |
| + SparseVLM | 320 | 1872 s (1.22×) | 644 s (1.65×) | 2.5 T | 78.5 |
| **+ Ours** | 320 | **921 s (2.48×)** | **360 s (2.95×)** | **1.2 T** | **84.8** |

## 5 CONCLUSION AND LIMITATIONS

In this paper, we proposed VisionTrim, a unified training-free framework for MLLM acceleration through comprehensive vision token compression. We presented two effective plug-and-play modules that accelerated both vision encoding and LLM decoding stages. By integrating the DVTS module, which selects tokens based on global semantics and local spatial continuity, with the TGVC module, which performs text-guided visual token complement, our approach consistently surpassed previous methods across various reduction ratios in both image and video understanding tasks.
**Limitations.** Although VisionTrim achieves 98.8% of the original performance with an 88.9% reduction ratio in token count without additional training costs, it is not entirely without loss. We are committed to advancing our research to further explore the redundancy of visual tokens and developing lossless methods to enhance the efficiency of visual understanding with MLLMs.

## ACKNOWLEDGEMENTS

This work is supported by the National Natural Science Foundation of China under Grant No.62376244. It is also supported by the Information Technology Center and State Key Lab of CAD&CG, Zhejiang University.

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

## TECHNICAL APPENDICES AND SUPPLEMENTARY MATERIAL

In this part, we provide additional details and more experimental results on our approach. The supplementary material is structured as follows:

- § A: Individual case performance analysis;
- § C: Additional implementation details;
- § D: More experimental results;
- § E: Broader impacts;
- § F: Asset license and consent.

## A  INDIVIDUAL CASE PERFORMANCE ANALYSIS

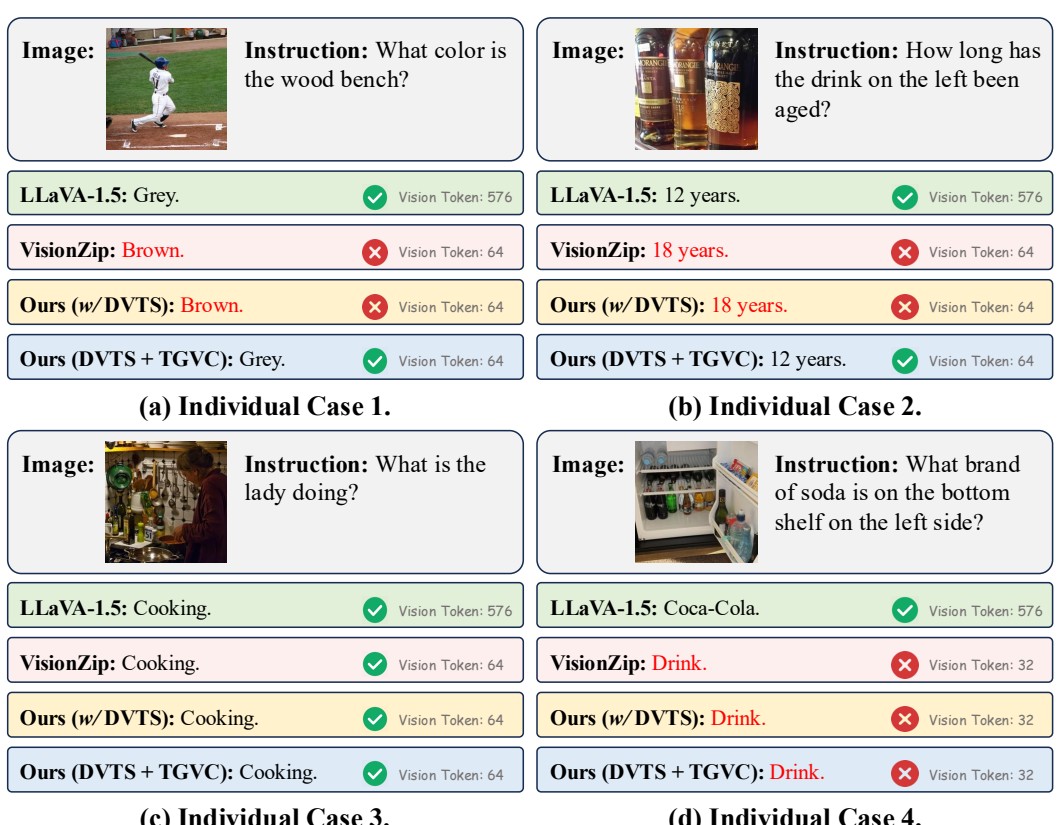

**(a) Individual Case 1.**  **(b) Individual Case 2.**

**(c) Individual Case 3.**  **(d) Individual Case 4.**

Figure 7: Visualized individual-case performance analysis concerning the phenomenon of knowledge boundary drift. Outputs highlighted in red indicate factual errors.

During the discussion phase, we conducted a detailed case study under an 88.9% token reduction ratio on LLaVA-1.5-7B, progressively applying the DVTS and TGVC modules, as shown in Figure 7. We observed an interesting pattern from the failure cases: when using only DVTS, knowledge boundary drift occurs in certain cases—i.e., the original model gives a correct answer, but after token pruning, the response becomes incorrect. However, after incorporating the TGVC module, the discarded tokens from DVTS are leveraged, and text-related visual tokens are explicitly complemented. This prevents the loss of critical information in the input to the model, thereby correcting the responses and mitigating knowledge boundary drift to some extent.

For example, in Figure 7(a) and (b), the original model provides correct answers, but applying VisionZip or the DVTS module alone leads to errors due to the loss of key visual cues. In contrast, combining DVTS with TGVC enables the model to recover the correct responses, highlighting TGVC's role in restoring essential text-related tokens. We attribute this behavior to VisionTrim's two-stage

Table 10: Additional performance-efficiency experiments on different components of VisionTrim in LLaVA-1.5-7B (under an 88.9% token reduction ratio).

| Methods | #Tokens | Total Inference Time ↓ | Prefill Time ↓ | Accuracy ↑ |
|---|---|---|---|---|
| Vanilla | 576 | 1303 s (1.00×) | 494 s (1.00×) | 85.9 |
| *w/* DVTS (only) | 64 | **670 s (1.94×)** | **227 s (2.18×)** | 84.6 |
| *w/* TGVC (only) | 64 | 741 s (1.76×) | 263 s (1.88×) | 85.0 |
| DVTS + TGVC (full) | 64 | 685 s (1.90×) | 235 s (2.10×) | **86.2** |

token compression process: DVTS first extracts dominant visual tokens via a dual-attention mechanism that captures global semantics and local spatial continuity, while TGVC complements discarded tokens using text guidance to mitigate the potential loss of critical visual information. Although DVTS pruning may remove some visual information, TGVC clusters and merges these tokens to preserve critical text-related features. This design allows VisionTrim to maintain the original model's performance while minimizing knowledge boundary drift.

In future work, we plan to perform targeted, case-driven tests on critical instances prior to deployment, including rigorous stability and reliability assessments for applications where stability is paramount, such as AI-based disease diagnosis. This will ensure that models processed with VisionTrim are reliable and trustworthy for real-world deployment.

## B  THE USE OF LARGE LANGUAGE MODELS (LLMS)

During the preparation of this paper, we utilized large language models (LLMs), specifically OpenAI's GPT-4 (Achiam et al., 2023), to assist with grammar refinement, sentence restructuring, and enhancing the overall readability of the manuscript. All scientific contributions, ideas, and evaluations presented in this work were independently developed and validated by the authors, without any involvement of LLMs.

## C  ADDITIONAL IMPLEMENTATION DETAILS

### C.1  COMPUTATIONAL BUDGET ESTIMATION

In a typical MLLM framework (Liu et al., 2023; 2024a; Zhu et al., 2023), the LLM decoder utilizes the causal self-attention mechanism (Vaswani et al., 2017), where each token attends only to the past tokens. The self-attention matrix $A \in \mathbb{R}^{L \times L}$, where $L$ is the length of the input sequence, is computed to model global dependence relationships among tokens as follows:

$$A = \text{Attention}(Q, K) = \text{Softmax}\left(Q \cdot K^{\text{T}} / \sqrt{d_k}\right), \tag{13}$$

where $Q \in \mathbb{R}^{L \times D}$ and $K \in \mathbb{R}^{L \times D}$ represent the query and key matrices, respectively, and the scalar $d_k$ is the dimension of these matrices. The computational complexity of each decoder layer includes the computation of the multi-head attention (MHA) mechanism and feed-forward network (FFN) module. For the entire LLM, the computational metric FLOPs after the $K$-*th* layer is estimated as:

$$\text{FLOPs}_{0:K-1} = K \times (4nd^2 + 2n^2d + 2ndm), \tag{14}$$

where $n$ denotes the token number, $d$ is the hidden state size, and $m$ is the intermediate size of FFN.

Therefore, the computational cost of MLLM exhibits a quadratic increase with respect to the quantity of input tokens. This highlights the significance of reducing the number of input tokens, particularly visual tokens (normally 20 times more numerous than system and question prompt tokens), while preserving the model's pioneering performance.

### C.2  THEORETICAL COMPUTATION REDUCTION

In analyzing computational efficiency, we focus on the FLOPs associated with visual tokens in the MLLMs. Here, $N$ denotes the initial number of visual tokens, and $K + R$ represents the number

Table 11: Token number settings for VisionTrim in LLaVA-1.5.

| LLaVA-1.5 | Retain 32 tokens | | Retain 64 tokens | | Retain 128 tokens | | Retain 192 tokens | |
|---|---|---|---|---|---|---|---|---|
| | DVTS module | TGVC module | DVTS module | TGVC module | DVTS module | TGVC module | DVTS module | TGVC module |
| **#Tokens** | 24 | 8 | 48 | 16 | 96 | 32 | 144 | 48 |

Table 12: Token number settings for VisionTrim in LLaVA-NeXT.

| LLaVA-NeXT | Retain 80 tokens | | Retain 160 tokens | | Retain 320 tokens | | Retain 640 tokens | |
|---|---|---|---|---|---|---|---|---|
| | DVTS module | TGVC module | DVTS module | TGVC module | DVTS module | TGVC module | DVTS module | TGVC module |
| **#Tokens** | 70 | 10 | 140 | 20 | 280 | 40 | 560 | 80 |

of tokens retained after our TGVC module ($K$ tokens for the main visual feature and $R$ tokens for the complement). For a single transformer layer with hidden size $d$ and intermediate size $m$, the theoretical FLOPs reduction ratio $F$, given a token reduction rate $\gamma = (K + R)/N$, is calculated as:

$$F = 1 - \frac{8\gamma N d^2 + 4(\gamma N)^2 d + 6\gamma N dm}{8N d^2 + 4N^2 d + 6N dm}. \tag{15}$$

The theoretical reduction formula above clearly illustrates that the computational cost of LLMs scales quadratically with the number of input visual tokens. This observation further underscores the importance of pruning and compressing redundant visual tokens during the forward propagation process of MLLMs.

## D    MORE EXPERIMENTAL RESULTS

### D.1    TOKEN NUMBER CONFIGURATIONS

In our method, the DVTS module captures the primary visual information, while the TGVC module utilizes the textual information to further achieve vision complementarity. Here, we provide our detailed token count configurations in the DVTS and TGVC modules, for normal resolution and high-resolution image-based MLLMs and video-based MLLMs.

For LLaVA-1.5 (Liu et al., 2024a), we detail our four configurations: 32 tokens, 64 tokens, 128 tokens, and 192 tokens, as summarized in Table 11. Regarding LLaVA-NeXT (Liu et al., 2024b), which is capable of handling high-resolution images, we specify the token counts as shown in Table 12. In the case of Video-LLaVA (Lin et al., 2023), which processes 8 frames per video with 256 tokens per frame, we perform token pruning to reduce the count to 17 tokens per frame, resulting in a total of 136 tokens for the entire video.

### D.2    MORE QUANTITATIVE RESULTS

### D.2.1    EXTREME TOKEN COUNT SETTING

Ultra-low token compression presents a significant challenge. To evaluate the effectiveness of our proposed method, we present quantitative experimental results under extreme token count settings. As shown in Table 13, VisionTrim is applied directly during inference without the need for additional training, while VisionTrim‡ involves efficient fine-tuning of the cross-modality attention module in TGVC and the projector, using only 1/10 of the LLaVA-1.5 training dataset. Notably, even with just **16 tokens**, VisionTrim retains 90% of the original performance in a training-free manner, demonstrating the robustness and efficacy of our approach.

Furthermore, we explore the extreme case of retaining only **1 token** resulting in a 99.8% reduction in token count. Surprisingly, VisionTrim still preserves approximately 82.8% of the original performance without any additional training. When fine-tuned with only 1/10 of the LLaVA-1.5 training dataset, VisionTrim‡ achieves 86.5% of the original performance. We also observe that as the number of retained tokens decreases and the compression ratio increases, the performance gains from fine-tuning become more pronounced. These experimental results further emphasize the potential of our approach for more flexible deployment and broader applicability in real-world scenarios.

Table 13: Results of our VisionTrim across different configurations of token counts on LLaVA-1.5-7B.

| Method | GQA | MMB | MME | POPE | SQA | VQA$^{V2}$ | VQA$^{Text}$ | SEED | MMVet | VizWiz | Avg. |
|---|---|---|---|---|---|---|---|---|---|---|---|
| *Upper Bound, 576 Tokens* **(100%)** | | | | | | | | | | | |
| Vanilla | 61.9 | 64.7 | 1862 | 85.9 | 69.5 | 78.5 | 58.2 | 58.6 | 31.1 | 50.1 | 100% |
| | 100% | 100% | 100% | 100% | 100% | 100% | 100% | 100% | 100% | 100% | |
| *Retain Averaged 16 Tokens* (↓97.2%) | | | | | | | | | | | |
| VisionTrim | 51.5 | 57.7 | 1510 | 72.6 | 69.4 | 68.4 | 53.0 | 53.4 | 28.9 | 50.0 | 90.0% |
| | 83.2% | 89.2% | 81.1% | 84.5% | 99.9% | 87.1% | 91.1% | 91.1% | 92.9% | 99.8% | |
| VisionTrim ‡ | 55.0 | 61.3 | 1612 | 78.1 | 70.2 | 72.3 | 55.4 | 55.9 | 30.2 | 50.5 | 94.3% |
| | 88.9% | 94.7% | 86.6% | 90.9% | 101.0% | 92.1% | 95.2% | 95.4% | 97.1% | 100.8% | ↑ (4.3%) |
| *Retain Averaged 8 Tokens* (↓98.6%) | | | | | | | | | | | |
| VisionTrim | 49.7 | 53.0 | 1491 | 70.1 | 68.2 | 64.4 | 52.7 | 53.3 | 27.9 | 49.5 | 87.4% |
| | 80.3% | 81.9% | 80.1% | 81.6% | 98.1% | 82.0% | 90.5% | 90.9% | 89.8% | 98.8% | |
| VisionTrim ‡ | 51.7 | 58.3 | 1525 | 75.2 | 69.9 | 70.2 | 55.3 | 54.4 | 30.4 | 49.9 | 91.8% |
| | 83.5% | 90.1% | 81.9% | 87.5% | 100.6% | 89.4% | 95.0% | 92.8% | 97.7% | 99.6% | ↑ (4.4%) |
| *Retain Averaged 4 Tokens* (↓99.3%) | | | | | | | | | | | |
| VisionTrim | 46.9 | 47.0 | 1460 | 64.0 | 67.8 | 66.6 | 51.6 | 52.2 | 25.7 | 50.5 | 84.5% |
| | 75.8% | 72.5% | 78.4% | 74.5% | 97.6% | 84.9% | 88.6% | 89.1% | 82.5% | 100.8% | |
| VisionTrim ‡ | 48.9 | 48.2 | 1496 | 65.4 | 68.6 | 72.5 | 52.0 | 53.6 | 29.6 | 51.4 | 88.0% |
| | 79.0% | 74.5% | 80.3% | 76.1% | 98.7% | 92.4% | 89.3% | 91.5% | 95.2% | 102.6% | ↑ (3.5%) |
| *Retain Averaged 1 Tokens* (↓99.8%) | | | | | | | | | | | |
| VisionTrim | 45.3 | 46.0 | 1441 | 64.1 | 66.7 | 65.7 | 50.4 | 51.7 | 24.5 | 49.4 | 82.8% |
| | 73.2% | 71.1% | 77.4% | 74.6% | 96.0% | 83.7% | 86.6% | 88.2% | 78.7% | 98.7% | |
| VisionTrim ‡ | 47.3 | 47.4 | 1435 | 64.6 | 68.5 | 70.2 | 51.3 | 53.3 | 29.3 | 50.7 | 86.5% |
| | 76.4% | 73.3% | 77.1% | 75.2% | 98.6% | 89.4% | 88.1% | 91.0% | 94.2% | 101.2% | ↑ (3.7%) |

Table 14: Performance comparison with other methods on InternVL2-2B (70.0% token reduction ratio).

| Method | OCRBench | OK-VQA | SQA | POPE | MMMU |
|---|---|---|---|---|---|
| Vanilla | 75.6 | 43.6 | 94.2 | 88.4 | 34.5 |
| FastV | 68.8 | 43.3 | 93.7 | 87.9 | 33.7 |
| TopV | 72.9 | 43.0 | 94.2 | 88.0 | 34.6 |
| Ours | 75.0 | 43.8 | 94.0 | 88.7 | 35.2 |

Table 15: Ablation experiment of different inserted LLM layers across various datasets on LLaVA-1.5-7B ($\alpha_1$ is set to 50%).

| Settings | GQA | MMB | POPE |
|---|---|---|---|
| $k = 2, \alpha_2 = 77.3\%$ | 60.4 | **64.2** | **86.2** |
| $k = 8, \alpha_2 = 66.7\%$ | 60.2 | 64.0 | 86.0 |
| $k = 12, \alpha_2 = 60.0\%$ | **60.5** | 63.8 | 85.5 |
| $k = 16, \alpha_2 = 50.0\%$ | 58.4 | 62.9 | 84.4 |
| $k = 20, \alpha_2 = 33.3\%$ | 57.3 | 63.3 | 84.1 |
| $k = 24, \alpha_2 = 0.0\%$ | 56.5 | 60.5 | 83.2 |

### D.2.2 ADDITIONAL TASK GENERALIZATION

In addition to the evaluation on classical MLLM benchmarks presented in the manuscript, we conduct more comprehensive experiments across various tasks to assess the generalization of VisionTrim. These additional evaluations encompass **generative image captioning** (as detailed in Table 17), **fine-grained visual classification** (as shown in Table 18), and **multi-modal retrieval** (as presented in Table 19). VisionTrim exhibits consistently superior performance across all benchmarks, effectively preserving or enhancing the original model's capabilities while outperforming existing SOTA methods by a significant margin. Notably, it achieves a marked reduction in inference time alongside a considerable boost in model throughput, thus advancing the practical deployment of MLLMs in real-world scenarios.

Table 16: Experiment results of deploying VisionTrim on LLaVA-OneVision-7B and Qwen2-VL-7B over single-/multi-image and video benchmarks. For LLaVA-OneVision, we utilize 243 tokens per image (down from the original 729 tokens) and 66 tokens per video frame (reduced from the original 196 tokens). Similarly, for Qwen2-VL, approximately 1/3 of the original tokens are used.

| Model | Single-image Benchmarks | | | | Multi-image Benchmarks | | | Video-based Benchmarks | | | |
|---|---|---|---|---|---|---|---|---|---|---|---|
| | MMMU | MMStar | MMBench | MMVet | MuirBench | Mantis | Q-Bench | Video-MME$_{(wo/w\ subs)}$ | PerceptionTest | Egoschema | MV-Bench |
| LLaVA-OneVision | 48.8 | 61.7 | 80.8 | 57.5 | 41.8 | 64.2 | 74.4 | 58.2 / 61.5 | 57.1 | 60.1 | 56.7 |
| *w/* VisionTrim | 48.6 (↓ 0.2) | 61.6 (↓ 0.1) | 80.5 (↓ 0.3) | 59.0 (↑ 1.5) | 41.4 (↓ 0.4) | 65.4 (↑ 1.2) | 74.3 (↓ 0.1) | 59.5 (↑ 1.3) / 61.4 (↓ 0.1) | 57.0 (↓ 0.1) | 62.3 (↑ 2.2) | 56.5 (↓ 0.2) |
| Qwen2-VL | 54.1 | 60.7 | 80.7 | 62.0 | – | – | – | 63.3 / 69.0 | 62.3 | 66.7 | 67.0 |
| *w/* VisionTrim | 53.8 (↓ 0.3) | 60.6 (↓ 0.1) | 82.8 (↑ 2.1) | 61.8 (↓ 0.2) | – | – | – | 63.1 (↓ 0.2) / 68.9 (↓ 0.1) | 62.0 (↓ 0.3) | 66.5 (↓ 0.2) | 68.8 (↑ 1.8) |

Table 17: Evaluation on the COCO Caption dataset with BLIP2.

| Method | BLEU@4 | CIDEr | FLOPs↓/Throughput↑ |
|---|---|---|---|
| BLIP2 | 42.8 | 145.6 | 1379.3 / 28.5 |
| Turbo | 42.1 | 142.2 | 989.7 / 50.9 |
| Ours | **43.2** | **145.4** | **752.4 / 67.0** |

Table 18: Evaluation on Referring Object Classification (ROC) Task with LLaVA-1.5-7B.

| Method | Box | Mask | Scribble | Point |
|---|---|---|---|---|
| Vanilla | 55.4 | 56.2 | 54.7 | 53.0 |
| VisionZip | 40.4 | 42.2 | 36.8 | 37.2 |
| Ours | **55.1** | **54.8** | **55.2** | **50.5** |

Table 19: Evaluation on Multi-modal Retrieval task across the COCO dataset with BLIP2.

| Method | Retrieval R@1 / R@5 | | FLOPs↓/Throughput↑ | △ |
|---|---|---|---|---|
| | Image-to-Text | Text-to-Image | | |
| BLIP2 | 85.4 / 97.0 | 68.3 / 87.7 | 717.5 / 10.8 | – |
| Turbo | 84.2 / 96.2 | 67.1 / 87.1 | 396.5 / 18.6 | 44.9% / 72.2% |
| Ours | **85.1 / 97.0** | **68.0 / 87.5** | **265.2 / 27.5** | **63.0% / 154.6%** |

### D.2.3 MORE EFFICIENCY ANALYSIS

We conduct further experiments incorporating additional efficiency metrics related to hardware-specific optimizations, as shown in Table 20. These experiments are performed on an NVIDIA A40 GPU, aiming to simulate real-world deployment scenarios. The additional efficiency metrics include **prefilling time**, **total inference time**, and **memory usage**. The results reveal that VisionTrim significantly accelerates MLLMs across various metrics, outperforming the previous SOTA method, VisionZip, by a substantial margin in both efficiency and accuracy. Notably, one particularly exciting finding is that with VisionTrim using 320 tokens, the 13B model achieves faster inference than the 7B model while maintaining superior performance.

### D.2.4 VARIOUS SIZES AND MODEL FAMILIES

In the main paper, we provided evaluation experiments on Qwen2-VL-7B and Qwen2.5-VL-7B across single-/multi-image and video benchmarks. Additionally, due to the page limit, we conduct more efficiency analysis on LLaVA-NeXT-13B in Table 20. Here we conducted more evaluation experiments comparing our VisionTrim against other methods on InternVL2-2B (Chen et al., 2024b), as shown in Table 14, and LLaVA-OneVision-7B (Li et al., 2024a), as shown in Table 16. Across different MLLMs of various sizes and model families, VisionTrim consistently maintains strong performance and efficiency, and also significantly outperforms existing baselines across multiple benchmarks, even under substantial token compression. This further highlights VisionTrim's superior ability to preserve critical visual details through our proposed token pruning strategy and modules.

### D.3 MORE ABLATION STUDY

### D.3.1 ABLATION STUDIES ON LAYER SELECTION.

In our default settings and the provided experiments, the proposed modules are inserted only between the second and third layers of the LLM, as presented in Table 26. To further clarify the implementation details, we conducted additional ablation experiments with different LLM layer selections, as shown in Table 15. Here, $k$ denotes the inserted LLM layer, and $\alpha$ represents the token retention rate after the inserted layer. To maintain the same overall token retention ratio of 75.0%

Table 20: More efficiency analysis on LLaVA-NeXT-7B/13B.

| Method | #Tokens | Prefilling Time (ms)↓ | Total Inference Time (s)↓ | Memory Usage (MB)↓ | Avg. Acc↑ |
|---|---|---|---|---|---|
| Vanilla-7B | 2880 | 112.4 | 2080 | 20874 | 96.2% |
| VisionZip-7B | 160 | 39.8 (3.2×) | 1224 (2.1×) | 17012 (-18.5%) | 91.3% |
| Ours-7B | 160 | **15.2 (7.4×)** | **547 (3.8×)** | **12482 (-40.2%)** | **94.8%** |
| Vanilla-13B | 2880 | 156.5 | 2814 | 36840 | 100% |
| VisionZip-13B | 320 | 57.3 (2.7×) | 1585 (1.8×) | 30862 (-16.2%) | 93.3% |
| Ours-13B | 320 | **28.0 (5.6×)** | **826 (3.4×)** | **25130 (-31.8%)** | **97.4%** |

Table 21: Evaluation results on OCR-Heavy datasets with LLaVA-1.5-13B (32 tokens).

| Method | OCRBench | OCRVQA | DocVQA | TextVQA | Avg. | Inference Time (s) |
|---|---|---|---|---|---|---|
| Vanilla | 331 | 54.9 | 45.2 | 59.4 | 100% | 2394 |
| VisionZip | 165 | 42.3 | 31.0 | 53.6 | 71.4% | 1096 |
| Ours | **297** | **55.2** | **44.3** | **57.6** | **96.3%** | **704** |

Table 22: Comparison results on Video-LLaVA-7B between our VisionTrim for inter-frame token compression and the uniform sampling method (136 tokens).

| Method | TGIF (Acc) | MSVD (Acc) | MSRVTT (acc) | ActivityNet (Acc) |
|---|---|---|---|---|
| Vanilla | 47.1 | 69.8 | 56.7 | 43.1 |
| Uniform Sampling | 45.2 | 68.6 | 54.9 | 43.5 |
| Ours | **47.4** | **69.3** | **55.8** | **46.0** |

throughout the full LLM decoding stage, we adaptively adjust $\alpha$ based on the inserted layer $k$. We observed that inserting the proposed modules into the shallow layers typically results in better performance.

### D.3.2 TRADE-OFFS FOR OCR-HEAVY TASKS

To strengthen the practical applicability of VisionTrim, we further include experiments on OCR-heavy benchmarks, as shown in Table 21. The results indicate that VisionTrim surpasses the previous SOTA method VisionZip, delivering 96.3% of the original MLLM performance while achieving 3.4× inference speedup. This is accomplished with a 94.4% token reduction ratio, effectively balancing accuracy and inference efficiency.

### D.3.3 INTER-FRAME TOKEN COMPRESSION FOR VIDEO-BASED MLLMS

Most video-based MLLMs often rely on uniform sampling of a fixed number of frames to achieve video understanding, which overlooks inter-frame token correlations. We leverage our VisionTrim to address temporal redundancy in videos through inter-frame token compression. Specifically, inspired by Chat-UniVi (Jin et al., 2024a) and LongVU (Shen et al., 2024), we first calculate the average similarity $Sim^i$ for each frame relative to all others using visual features extracted from the vision encoder like CLIP (Radford et al., 2021). We then select top-$K$ frames with the lowest $Sim^i$ values as cluster centers and assign the remaining $N - K$ frames to their nearest cluster. Similar to per-frame token merging, we consolidate non-essential frames in each cluster by merging $K$-nearest neighbor tokens across frame features. The comparison results are reported in Table 22.

### D.3.4 COMPARISON WITH DIRECT TEXT-GUIDED METHODS

Direct text-guided methods (Ju et al., 2024; Shi et al., 2024) can hinder multi-round interactions in MLLMs and potentially lead to hallucinations. Our two-stage approach addresses these issues by initially preserving core visual tokens through the DVTS module and subsequently aggregating text-related tokens using the TGVC module. As shown in Table 23, VisionTrim outperforms the direct text-guided method, particularly in benchmarks related to hallucinations and OCR-related tasks.

### D.3.5 Comparison with Random Pruning

We also provide the experiments to compare VisionTrim with indiscriminate random pruning, as shown in Table 24. In contrast to the random pruning, our approach achieves significant performance improvement across all benchmarks.

### D.3.6 Generalization to Non-CLIP Vision Encoders

To further assess the generalizability of the VisionTrim framework, we test it on non-CLIP vision encoders, such as DINOv2 (Oquab et al., 2023). Specifically, we first replace the CLIP vision encoder in LLaVA with DINOv2 and fine-tune LLaVA accordingly. Subsequently, we apply VisionTrim to the fine-tuned MLLM to evaluate its generalizability. The experimental results are presented in Table 25.

### D.3.7 Hyperparameter Configurations for Reproduction

In our proposed DVTS and TGVC modules, we maintain consistency in hyperparameters across different MLLMs and benchmarks. To facilitate the reproduction of our VisionTrim's experimental results, we provide detailed hyperparameter settings in Table 26. For MLLM parameters, we use the default settings of each base MLLM, as our method is applied during inference on pre-trained models.

### D.3.8 Different Components in VisionTirm

We conduct additional performance–efficiency experiments on different components added to the final model, as shown in Table 10. We observe that using only the DVTS module further improves inference efficiency, but leads to a larger performance drop because text-related information is not considered. In contrast, using only the TGVC module preserves performance more effectively, but yields a smaller improvement in inference efficiency due to the increased token-level similarity computations involved in clustering. By combining both modules, the final model achieves a better trade-off between performance and efficiency.

### D.4 More Case Studies

Figure 8 showcases several examples of VisionTrim's application in both image- and video-based understanding tasks. Unlike other methods that typically rely on a larger number of visual tokens, VisionTrim effectively captures visual details with a minimal token set. For instance, VisionTrim successfully distinguishes between different characters in an image and accurately identifies the license plate number on a bus.

In the context of video understanding, Video-ChatGPT (Maaz et al., 2023) utilizes a fixed number of visual token representations across the entire video, which limits its ability to generate comprehensive descriptions that capture temporal dynamics. Video-LLaVA (Lin et al., 2023) processes eight frames per video, allocating 256 tokens per frame. Nonetheless, this approach inherently contains substantial visual redundancy, potentially leading to hallucinations. For example, in the case of a soccer game video, Video-LLaVA misinterprets the player's footwork and interactions with defenders. In contrast, VisionTrim effectively retains crucial visual information while significantly reducing redundancy. Our approach not only enhances visual comprehension but also optimizes efficiency, making it a more practical and viable solution for efficient multimodal interaction in real-world applications.

### D.5 More Visualization Results

#### D.5.1 Token Redundancy in Vision Encoding Stage

MLLMs typically leverage the encoded features from the penultimate layer of the vision encoder. To further illustrate the redundancy within vision encoders, we present additional visualization results. As shown in Figure 9, only a small subset of tokens receives high attention to capture the key visual information, while the majority of visual tokens are allocated minimal attention and contribute little

Table 23: Experimental results comparing our VisionTrim with the direct text-guided method on LLaVA-1.5-13B (32 tokens).

| Method | POPE | MME | OCRBench | DocVQA | OCRVQA | TextVQA |
|---|---|---|---|---|---|---|
| Vanilla | 86.2 | 1875 | 331 | 45.2 | 54.9 | 59.4 |
| Direct Text Guidance | 59.5 | 1240 | 154 | 28.4 | 37.5 | 47.4 |
| Ours | **84.4** | **1685** | **297** | **44.3** | **55.2** | **57.6** |

Table 24: Comparison of VisionTrim *vs.* Random Pruning on LLaVA-1.5-7B (32 tokens).

| Method | GQA | MMB | POPE | SQA | VQA$^{V2}$ |
|---|---|---|---|---|---|
| Random Pruning | 40.8 | 51.4 | 62.8 | 60.5 | 56.7 |
| Ours | **55.3** | **60.2** | **77.5** | **70.1** | **72.6** |

Table 25: Evaluation results on LLaVA-1.5-7B using DINOv2 as the vision encoder (32 tokens).

| Method | SEED | MMB | MME | POPE | SQA |
|---|---|---|---|---|---|
| Vanilla | 60.6 | 64.2 | 1404 | 86.7 | 70.0 |
| Ours | 58.5 | 63.0 | 1368 | 83.4 | 70.7 |

Table 26: Detailed hyperparameter configurations in VisionTrim.

| Parameter | Value |
|---|---|
| Kernel-size in LTAM | $3 \times 3$ |
| $(w_1, w_2, w_3)$ in LTAM | (0.3,0.3,0.5) |
| Inserted layer in ViT (pre-LLM) | 23rd (total 24 layers) |
| Inserted layer in LLM (within-LLM) | 2nd (total 32 layers) |
| Token count retained in DVTS *vs.* TGVC | 3:1 |
| Token retention rate $\alpha_1$ in pre-LLM stage | 50% |
| Token retention rate $\alpha_2$ in within-LLM stage | Dynamically adjusted |

to the overall visual content. This observation underscores the significant redundancy among visual tokens, emphasizing the necessity for their compression.

### D.5.2  TOKEN REDUNDANCY IN LLM DECODING STAGE

To examine the token redundancy in the LLM decoding stage, we provide visualizations of the attention scores at the 16th and 32nd layers of the LLM as illustrated in Figure 10 and Figure 11. Consistent with the findings from the vision encoding stage, only a limited number of tokens receive high attention, while most visual tokens are largely ignored. This suggests that many of these less relevant visual tokens should be pruned during the LLM propagation phase.

### D.5.3  ANALYSIS OF LAYER-WISE ATTENTION DISTRIBUTION

Based on the observations discussed above, a natural question arises: What are the underlying causes of redundancy in visual tokens? In this section, we present a comprehensive analysis of the attention distribution of layers within both vision encoders and LLMs.

As illustrated in Figure 12, attention in the shallow layers is broadly distributed across the entire image. However, as the model progresses to the middle layers, attention rapidly consolidates around a smaller subset of tokens. In the deeper layers, it becomes increasingly concentrated on a limited number of dominant tokens, reaching its peak concentration by the 23rd layer, which is then utilized for visual token extraction for the LLM. A similar trend is observed during the LLM decoding phase. As shown in Figure 13, attention initially distributes relatively evenly across all visual tokens. However, as the number of layers increases, the LLM's attention progressively narrows to focus on only a few tokens, while neglecting the majority of visual tokens.

To assess the effectiveness of our proposed DVTS and TGVC modules, we visualize the attention maps across all 32 layers during the LLM decoding stage, both with and without VisionTrim. We set VisionTrim to reduce the vanilla 576 visual tokens to 128 tokens in the vision encoding stage before being input to the LLM. For comparison, we randomly select 128 visual tokens for the baseline LLM. As shown in Figure 14, the vanilla LLaVA-1.5-7B model exhibits the aforementioned redundancy in visual tokens, particularly after the shallow layers (*i.e.*, the first two layers), where attention is concentrated on only a small subset of tokens, largely disregarding the majority of visual tokens. In contrast, as depicted in Figure 15, VisionTrim enhances the cross-modal alignment be-

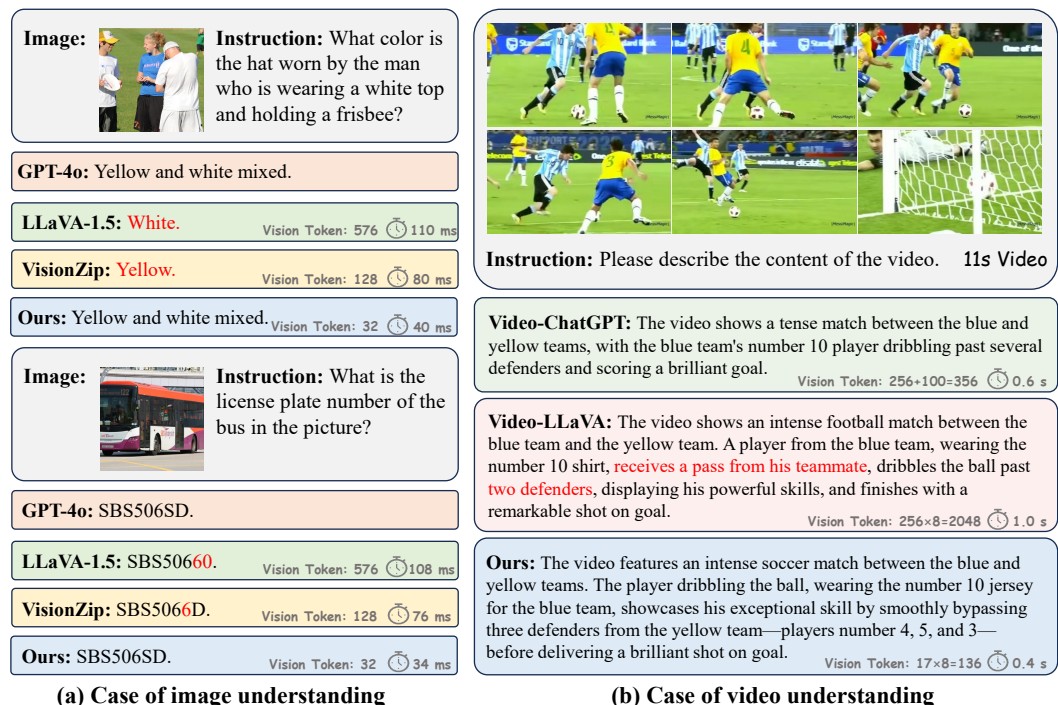

**(a) Case of image understanding**      **(b) Case of video understanding**

Figure 8: Visual examples showcasing VisionTrim's ability to accurately capture detailed visual information in both images and video. Outputs highlighted in red indicate factual errors.

Table 27: Open-source resources utilized in this paper.

| Name | License | URL |
|------|---------|-----|
| TextVQA Dataset | BSD License | https://textvqa.org/ |
| VQA Dataset | BSD License | https://github.com/GT-Vision-Lab/VQA |
| ScienceQA Dataset | MIT License | https://github.com/lupantech/ScienceQA |
| POPE Dataset | MIT License | https://github.com/AoiDragon/POPE |
| LLaVA-1.5 | Apache License 2.0 | https://github.com/haotian-liu/LLaVA |
| LLaVA-NeXT | Apache License 2.0 | https://github.com/LLaVA-VL/LLaVA-NeXT |
| Video-LLaVA | Apache License 2.0 | https://github.com/PKU-YuanGroup/Video-LLaVA |
| MMBench Dataset | Apache License 2.0 | https://github.com/open-compass/MMBench |
| MME | Apache License 2.0 | https://github.com/BradyFU/Awesome-Multimodal-Large-Language-Models |
| MM-Vet Dataset | Apache License 2.0 | https://github.com/yuweihao/MM-Vet |
| SEED-Bench Dataset | Apache License 2.0 | https://github.com/AoiDragon/POPE |
| COCO Dataset | Creative Commons Attribution 4.0 | https://cocodataset.org/#home |
| VizWiz Dataset | Creative Commons Attribution 4.0 | https://vizwiz.org/tasks-and-datasets/vqa/ |
| GQA Dataset | - - - | https://cs.stanford.edu/people/dorarad/gqa/index.html |
| OCR-VQA | - - - | https://ocr-vqa.github.io/ |

tween visual and textual tokens, enabling the LLM to more effectively focus on the retained image tokens while preserving the most salient visual information. This improvement not only maintains model performance but also significantly boosts inference speed and efficiency.

# E  BROADER IMPACTS

This paper introduces a training-free method, named VisionTrim, designed to accelerate Multimodal Large Language Models (MLLMs) through two plug-and-play modules. These modules effectively filter the essential visual tokens, maintaining both global semantics and local continuity, while leveraging textual information to guide token merging, thereby enhancing the visual-textual alignment. On the positive side, our approach has the potential to significantly benefit the efficient deployment of MLLMs for real-world image and video understanding tasks, offering a clear reduction in training and inference costs while maintaining competitive performance. However, due to the inherent robustness challenges of large multimodal models, some erroneous outputs may result in misinformation or safety concerns (Wei et al., 2023; 2024). To mitigate these risks, we recommend

implementing a stringent security protocol to address potential failures of our approach in practical multimodal applications.

## F  ASSET LICENSE AND CONSENT

We conduct an extensive evaluation across 10 widely-used image-based benchmarks and 4 popular video-based tasks to assess the multi-modal understanding and reasoning capability of our proposed VisionTrim. Additionally, we utilize 66K mixture instruction-following data (a subset of the original 665K LLaVA-1.5 dataset) for efficient instruction tuning. All datasets are publicly available and free for academic research. Table 27 lists the resources used in this research work along with their associated licenses.

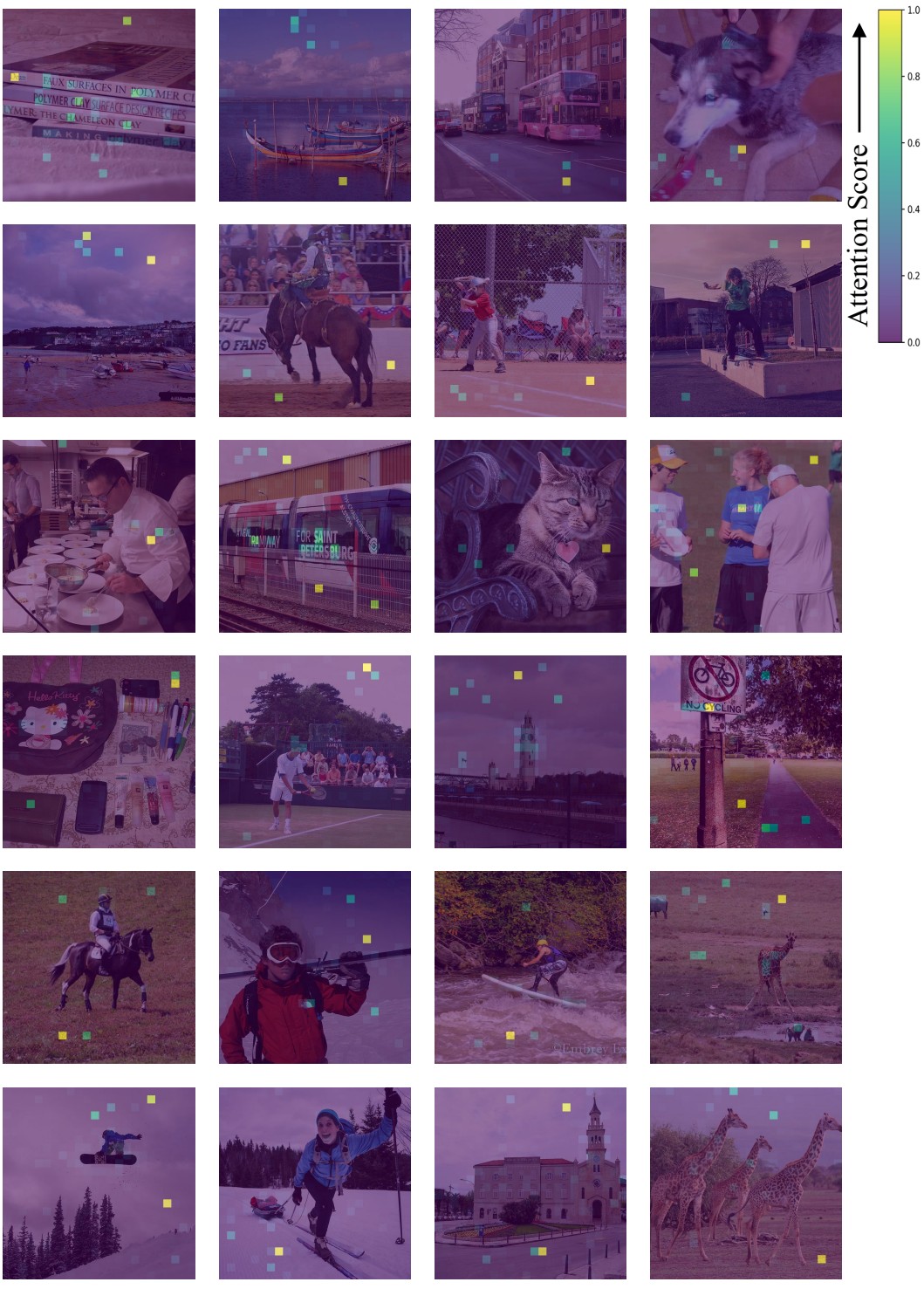

Figure 9: Visualization of redundancy in the penultimate layer of the vision encoder (CLIP Model).

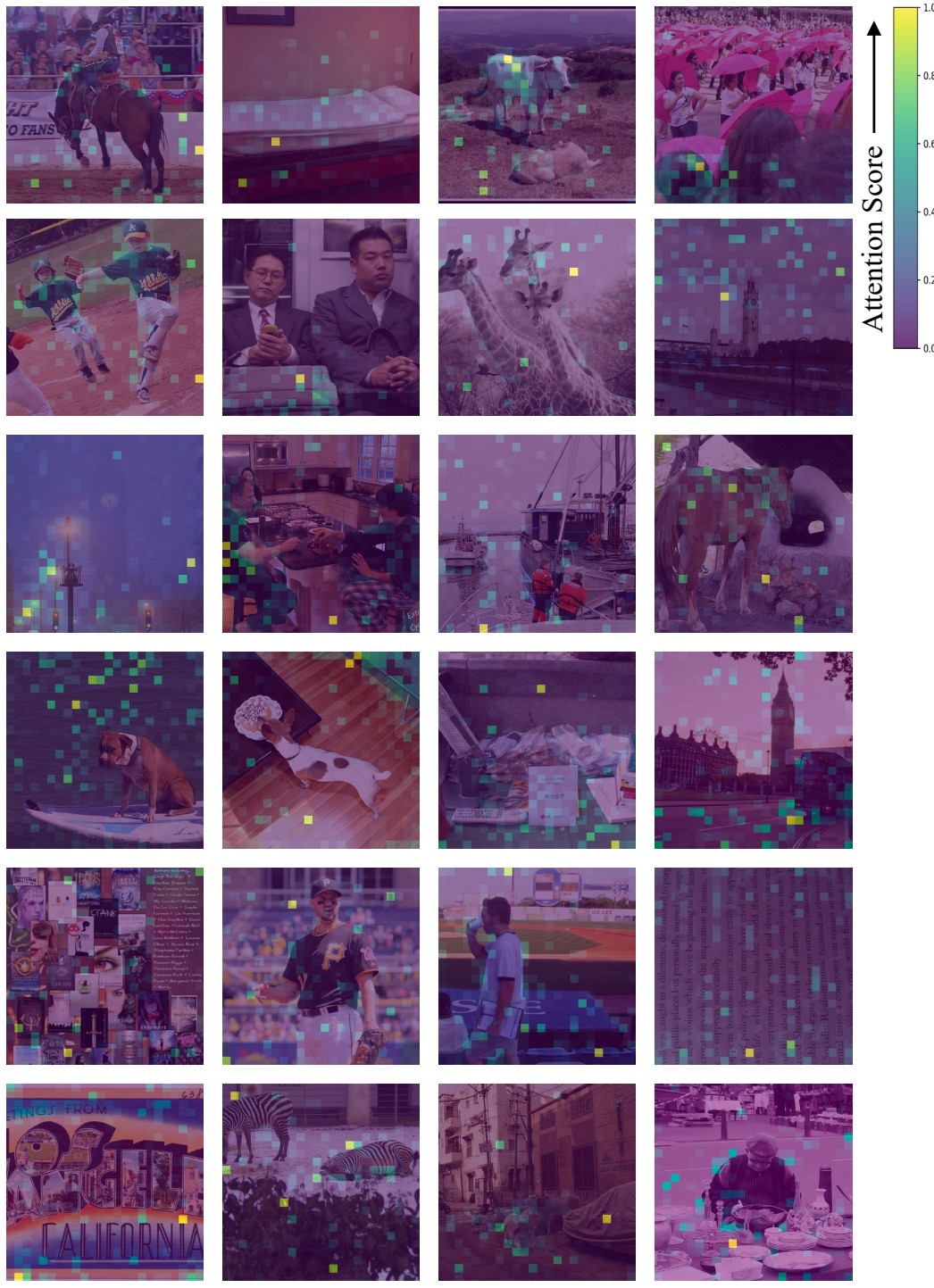

Figure 10: Visualization of redundancy in the 16th layer of LLM.

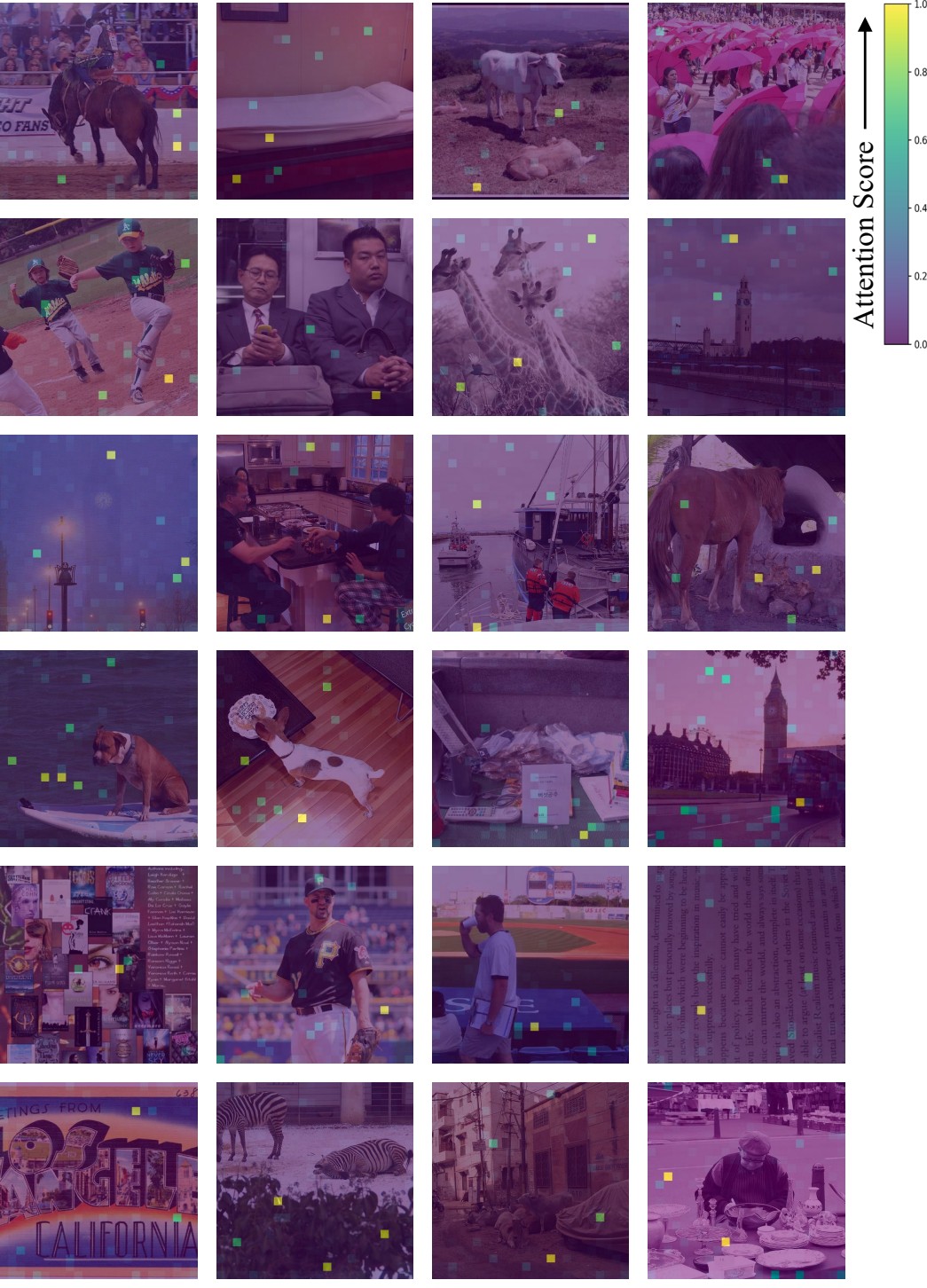

Figure 11: Visualization of redundancy in the 32nd (final) layer of LLM.

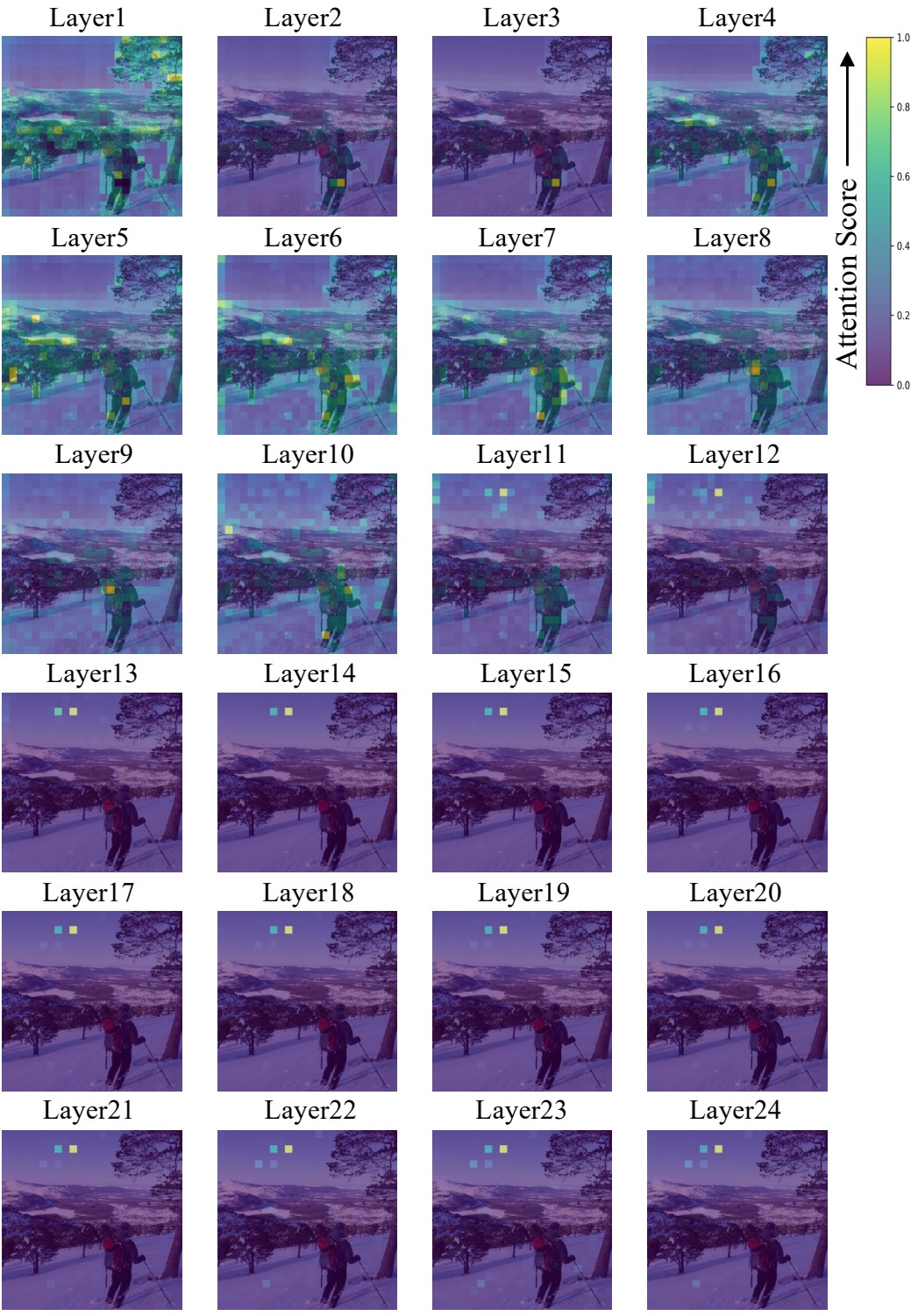

Figure 12: Visualization of attention distribution change in vision encoder (CLIP Model).

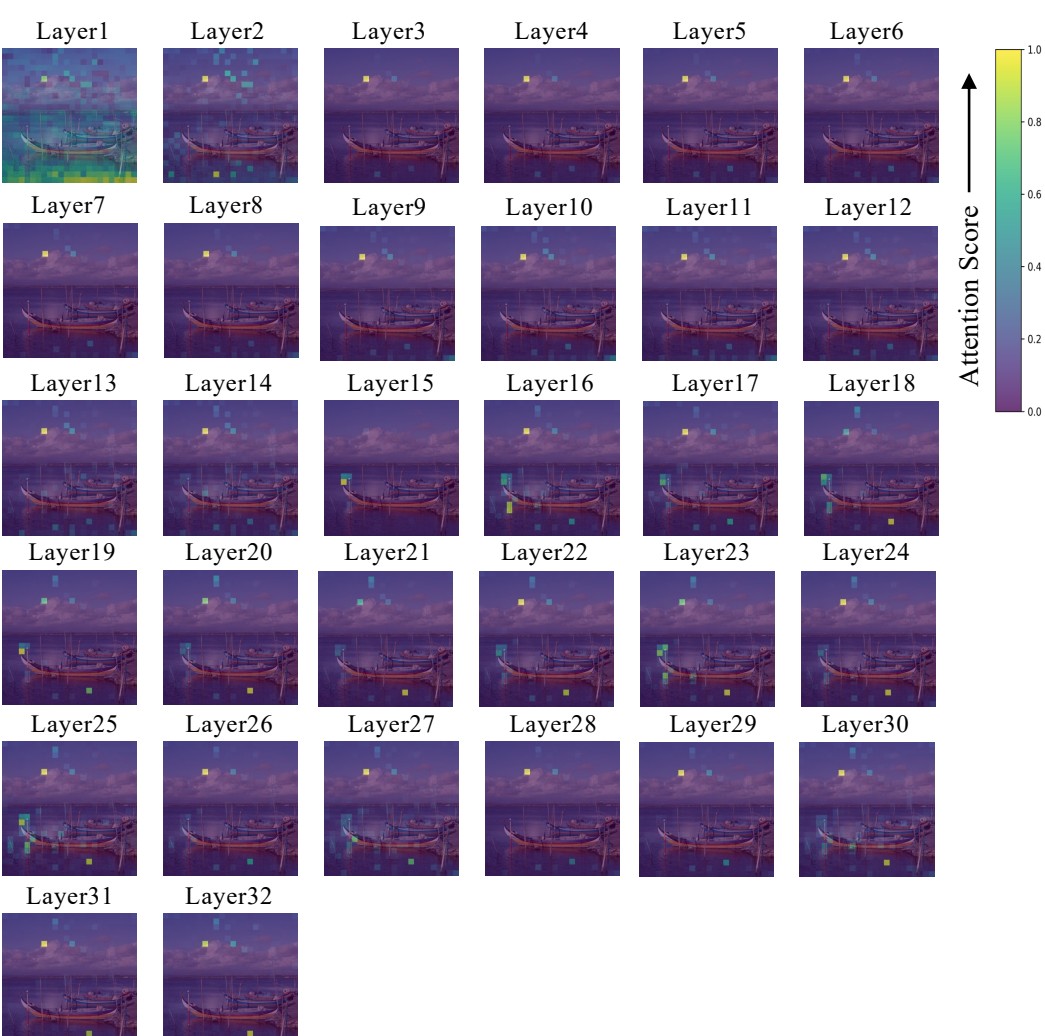

Figure 13: Visualization of changes in attention distribution across all 32 layers in LLM..

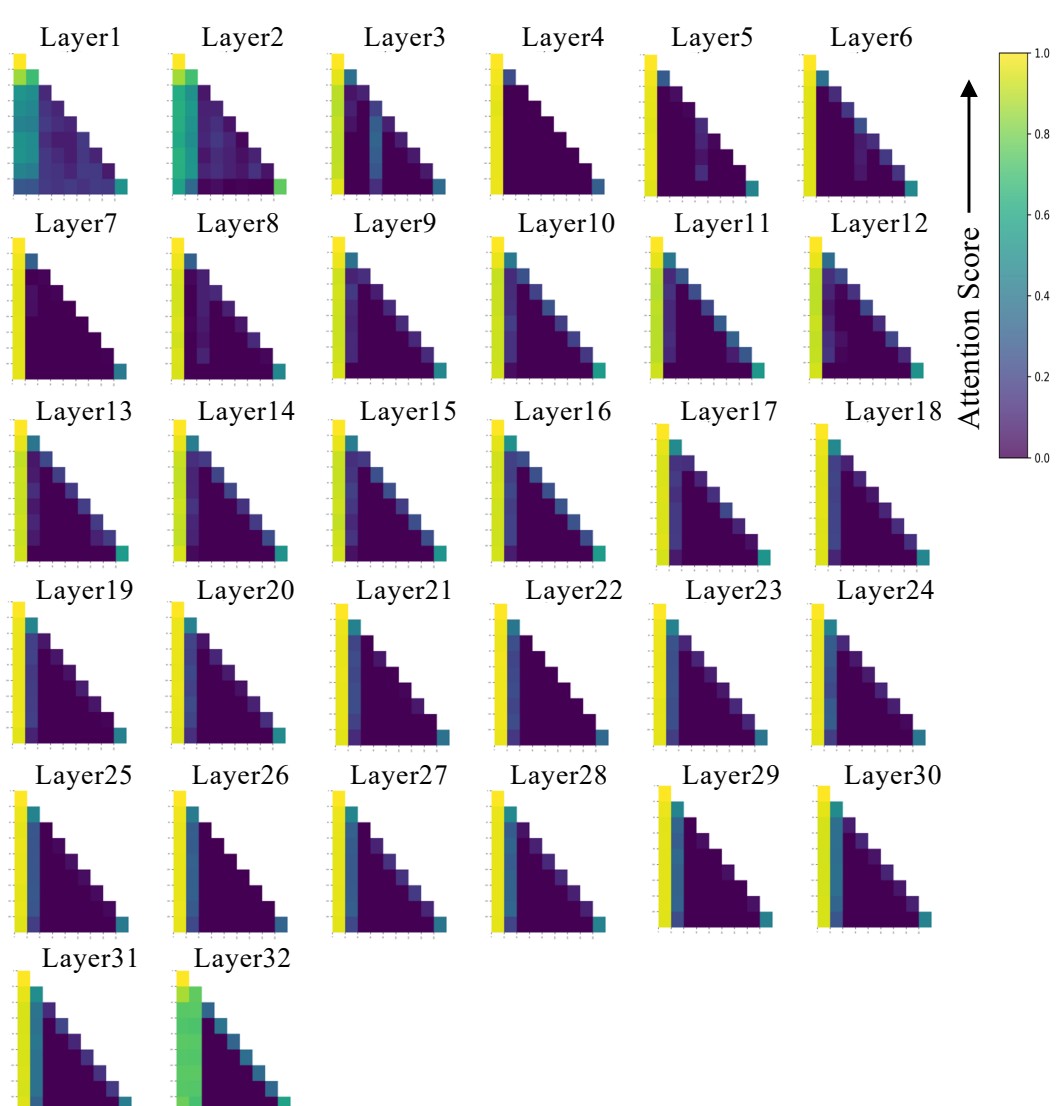

Figure 14: Visualization of attention maps across all 32 layers in vanilla LLM processing.

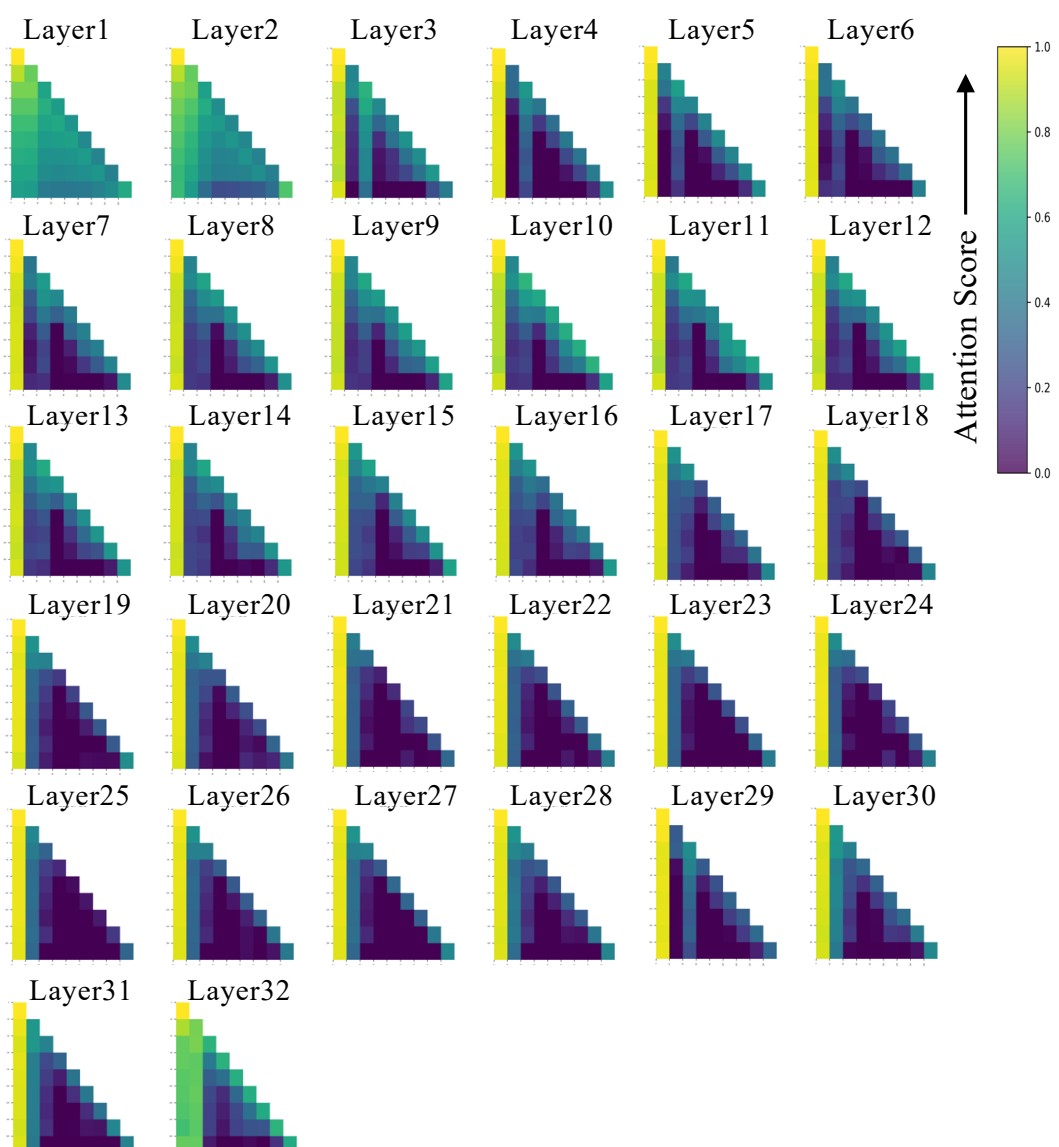

Figure 15: Visualization of attention maps across all 32 layers in LLM processing with our proposed VisionTrim.

