# OpenReview forum: "VisionTrim: Unified Vision Token Compression for Training-Free MLLM Acceleration"
_ICLR.cc/2026/Conference — ICLR 2026 Poster_

### Official Review · Reviewer_8p5d · 2025-10-25

**Soundness:** 3
**Presentation:** 2
**Contribution:** 3
**Rating:** 6
**Confidence:** 3

**Summary:**

This work proposes a training-free token pruning method for model acceleration. Specifically, the authors designed a two-stage plug-and-play module. First, DVTS uses a pre-trained CLIP-type encoder to prune vision tokens. Then, the TGVC module clusters and merge the pruning vision tokens for ensure semantically relevant details are preserved. Result shows that the proposed method can still perform well even when pruning 88% of the tokens.

**Strengths:**

The experiments are comprehensive, as the authors tested the proposed method on more than just a single model. Furthermore, the authors evaluated the use of a key-value cache. Diagrams and qualitative results are clear and intuitive. The findings of this paper are highly instructive for industrial applications, particularly the key proposal to tailor both the encoder and decoder.

**Weaknesses:**

1. Academic writing requires standardization. It appears that the author does not use citation formatting well. Most citations are in the form of "name et al (year)," which makes them difficult to read.

2. To my limit knowledge, token pruning or quantization methods can ensure that overall performance does not drop significantly. However, they may introduce knowledge boundary drift, meaning that answers that were previously correct may now be incorrect, and vice versa [1]. This possibility could have negative consequences in scenarios where critical questions must be answered correctly. If possible, could the authors conduct a simple analysis to see whether the proposed method leads to changes in individual case performance?

[1] Sun, Yizheng, et al. "Does Acceleration Cause Hidden Instability in Vision Language Models? Uncovering Instance-Level Divergence Through a Large-Scale Empirical Study." arXiv preprint arXiv:2503.06794 (2025).

**Questions:**

1. Does the proposed pulg-in module have the same weights in the encoder and decoder stages?
2. Are these components plugged into each layer?
3. Is top-k selection also mandatory during training? If so, how can author ensure it can be effectively generalized to different selected scales during inference?

---

> ### Author Response · Authors · 2025-11-20
> **Response to Reviewer 8p5d (1/2)**
>
> We sincerely thank Reviewer 8p5d for the careful reading, for recognizing the practical value of our findings, the breadth of our experiments, and the clarity of our diagrams and qualitative results. Below, we address each concern and describe the concrete revisions we have made.
>
> ---
>
> > ### **`Weakness 1`: An academic writing issue concerning the usage of citation formatting.**
>
> Thank you for your valuable feedback. We fully agree that citations in the format “Name et al. (Year)” can be difficult to read. However, **this citation style is enforced by the official ICLR2026 LaTeX template, so we are required to follow it.** After the rebuttal phase, we plan to notify the ICLR2026 committee about this issue and suggest updating the LaTeX template to adopt a more readable citation format.
>
> ---
>
> > ### **`Weakness 2`: Conduct a simple analysis to see whether the proposed method leads to changes in individual case performance.**
>
> We sincerely thank you for the valuable feedback. We carefully reviewed the paper [1] regarding knowledge boundary drift after token pruning. Following your recommendation, we conducted a detailed case analysis at an 88.9% token reduction ratio, progressively applying the DVTS and TGVC modules.
>
> We observed **an interesting phenomenon from the failure cases**: when using only DVTS, knowledge boundary drift occurs in certain cases—i.e., the original model provides a correct response, but after token pruning, the response becomes incorrect. However, after applying the TGVC module, the discarded tokens from DVTS are leveraged, and text-related visual tokens are explicitly complemented. This prevents the loss of critical information in the input to the model, thereby correcting the responses and mitigating knowledge boundary drift to some extent.
>
> For example, for the question `What color is the wood bench?` with an image depicting a baseball player hitting a ball on a brown field next to a gray wooden bench, the original model correctly outputs `Grey`. However, after the DVTS module alone, the model’s response shifts to `Brown` due to the loss of crucial visual cues about the bench. In contrast, with both DVTS and TGVC, the model’s response returns to the correct `Grey`, demonstrating TGVC’s ability to recover essential text-related tokens and reduce information loss. **Visualized examples of these QA cases are provided in `Appendix A`, under the “Individual Case Performance Analysis” section on page 14 of the revised manuscript.**
>
> We attribute this phenomenon to VisionTrim’s two-stage token compression process. First, the DVTS module extracts dominant visual tokens using a dual-attention mechanism that captures both global semantics and local spatial continuity. Next, the TGVC module applies text-based complementation to the discarded tokens, mitigating the loss of essential text-related visual cues. Although DVTS pruning may remove some visual information, TGVC clusters and merges these tokens to preserve critical text-related features. **This design allows VisionTrim to maintain the original model’s performance while minimizing knowledge boundary drift.**
>
> In future work, we plan to conduct targeted, case-driven tests on critical instances before deployment, including rigorous stability and reliability checks for stability-centered applications such as AI-based disease diagnosis. This will ensure that VisionTrim-processed models are truly faithful and trustworthy for real-world deployment. We will include the above analysis and visualizations in the main paper to further highlight the advantages of VisionTrim.
>
> ---
>
> *[1]  Sun, Yizheng, et al. "Does Acceleration Cause Hidden Instability in Vision Language Models? Uncovering Instance-Level Divergence Through a Large-Scale Empirical Study." arXiv preprint arXiv:2503.06794 (2025).*

---

> ### Author Response · Authors · 2025-11-20
> **Response to Reviewer 8p5d (2/2)**
>
> ---
>
> > ### **`Question 1`: Do the proposed plug-in modules have the same weights in the encoder and decoder stages?**
>
> For the proposed DVTS and TGVC modules, we use the same parameter configuration in both the vision encoder and the LLM decoder to facilitate practical deployment and usage. As shown in `Table 1`, we provide the detailed hyperparameters used in our experiments. **For all MLLMs involved in the experiments, the results were obtained using the same set of hyperparameters.** Additionally, we will make all the code and hyperparameters used in these modules publicly available to promote the practical application and reproducibility of VisionTrim in accelerating MLLMs.
>
> **Table 1. Detailed hyperparameter configurations used in VisionTrim.**
>
> | Parameter                                                    | Value                  |
> | ------------------------------------------------------------ | ---------------------- |
> | Inserted layer in ViT (pre-LLM)                              | 23rd (total 24 layers) |
> | Inserted layer in LLM (within-LLM)                           | 2nd (total 32 layers)  |
> | Kernel size in LTAM                                          | 3×3                    |
> | Retained token count $K$ in DVTS : Retained token count $R$ in TGVC | 3:1                    |
> | ($w_1$,$w_2$,$w_3$) in LTAM                                  | (0.3,0.3,0.1)          |
>
> ---
>
> > ### **`Question 2`: Are these components plugged into each layer?**
>
> In the vision encoder, the proposed components are inserted only between the penultimate and last layers, while in the LLM decoder, they are inserted only between the second and third layers, as shown in `Table 1`.
>
> ---
>
> > ### **`Question 3`: Is top-$K$ selection mandatory during training?**
>
> VisionTrim operates in a training-free manner, meaning it is applied exclusively during the inference stage and **does not interfere with the model's training process**. This design allows the underlying models to be trained in the usual way, **without any modifications or the need for fixed token selection during training**. By operating solely at inference, VisionTrim significantly boosts efficiency while preserving model performance, demonstrating strong generalizability to different selected scales and practical applicability in real-world MLLM deployments.
>
> ---
>
> We thank Reviewer 8p5d again for the insightful comments. We believe that the revisions outlined—an expanded analysis of the knowledge boundary drift phenomenon after token pruning and more detailed hyperparameter configurations—will significantly strengthen the paper. We sincerely hope our responses have adequately addressed your concerns and look forward to any further discussion.

---

> > ### Comment · Reviewer_8p5d · 2025-11-21
> >
> > Thank you for the comprehensive rebuttal. The clarifications provided effectively resolve my concerns. For Weakness 1, both citation formats (“Author et al., Year” and “Author et al. (Year)”) are acceptable depending on sentence context; maybe check other submit papers. As my initial overall assessment was positive, I will maintain my score.

---

> ### Author Response · Authors · 2025-11-21
> **Thank you for your supportive feedback!**
>
> Dear Reviewer 8p5d,
>
> We sincerely appreciate your careful review and for recognizing our efforts to effectively address your concerns. Your support and positive recommendation are truly encouraging, and we deeply value your constructive comments. Thank you once again for your time and insightful feedback!
>
> Best regards,
>
> The Authors

---

### Official Review · Reviewer_M569 · 2025-10-26

**Soundness:** 2
**Presentation:** 2
**Contribution:** 3
**Rating:** 4
**Confidence:** 4

**Summary:**

-   The paper introduces VisionTrim, a training-free framework for reducing vision tokens in MLLMs are a way of accelerating inference.
-   VisionTrim proposes two modules: Dominant Vision Token Selection (DVTS) and Text-Guided Vision Complement (TGVC).
	-   DVTS: Aims to preserve visual integrity during visual token compression by first identifying dominant visual tokens.
	-   TGVC: Utilize text prompts to cluster remaining vision tokens through CLIP embeddings. Each cluster is then merged.
-   Experimental results show that VisionTrim improves on existing methods by around 1-2% averaged across benchmarks.
-   Furthermore, there is also a significant reduction in CUDA Time, FLOPS, and KV cache memory usage.

**Strengths:**

-   The redundancy in vision tokens in MLLMs is definitely an area of concern.
-   Unlike existing methods, which compress vision tokens in either the vision encoder or the LLM decoder, VisionTrim proposes a unified framework that can be applied to both modules.
-   Experimental results show general improvement over existing methods by around 1-2% on average score across benchmarks, with more pronounced improvements at a lower number of vision tokens per image, showing its effectiveness at preserving visual information.
-   Paper is generally easy to read (except for some equations that lack elaborations). The diagrams and tables illustrating the methods are clear and well-drawn.

**Weaknesses:**

-   The paper positions itself as improving inference latency of MLLMs and reports on CUDA time for decoding, which is an odd metric to use for inference latency (most papers report end-to-end inference latency or time to first token). However, to my knowledge, clustering and pairwise similarity between tokens can be slow, given the high dimensions in both the vision token count and the embedding size. The authors did not mention whether clustering is performed on the CPU or GPU. The authors did not disclose the overall end-to-end latency; with only the CUDA time reported, I remain sceptical that this method is an effective approach at reducing inference latency.
-   The proposed VisionTrim is a unified framework that can be plugged into both the vision encoder and the LLM decoder. However, there is a lack of experiments exploring the configuration: what percentage of vision tokens are pruned in the vision encoder / LLM decoder, respectively? At what stages are they pruned? What are the performance/efficiency trade-offs in pruning in the vision encoder vs in the LLM decoder? The lack of details would make it difficult to reproduce experiments for future work.
-   Minor presentation issues: some equations, especially those in local spatial continuity, are not well explained. The intuition behind these could be further elaborated.

**Questions:**

-   The paper describes TGVC as a training-free plug-and-play module. However, Figure 3(a) shows a [fire] symbol over TGVC, while a [snowflake] symbol over other modules, such as the vision encoder and projector, which are not explained. It is unclear whether this was an intended design or an oversight. Could the authors clarify this? (Fig. 1 as well)

---

> ### Author Response · Authors · 2025-11-20
> **Response to Reviewer M569 (1/3)**
>
> We sincerely thank Reviewer M569 for the careful reading, for recognizing the novelty of our unified framework, the consistent performance gains across benchmarks, and the clarity of our diagrams and tables. Below, we respond to each concern and detail the corresponding revisions.
>
> ---
>
> > ### **`Weakness 1`: Provide a more comprehensive efficiency analysis of the overall end-to-end latency.**
>
> Thank you very much for your valuable suggestion. We have conducted additional efficiency experiments evaluating **both the overall end-to-end latency and the prefill time** on a single NVIDIA A100 GPU, as shown in `Table 1`. VisionTrim achieves a **2.48×** speedup in overall end-to-end latency and a **2.95×** speedup in prefill time on LLaVA-NeXT-7B, while maintaining robust or even improved performance, clearly surpassing existing methods in both accuracy and efficiency.
>
> We sincerely apologize for the confusion. In the TGVC module, clustering and similarity computations are executed on the GPU to enable parallel processing and accelerate computation. **The clustering step simply assigns tokens based on the precomputed similarities between the center visual token and the remaining visual tokens, introducing negligible overhead.** We measured that the token merging process in TGVC takes less than `1 ms` per sample on the POPE benchmark.
>
> **Table 1. Extended efficiency analysis on the POPE benchmark (under an 88.9% token reduction ratio).**
>
> | Method        | #Tokens | Total Inference Time ↓ | Prefill Time ↓   | FLOPs ↓  | Accuracy ↑ |
> | ------------- | ------- | ---------------------- | ---------------- |-------- | ---------- |
> | LLaVA-1.5-7B  | 576     | 1303s (1.00x)          | 494s (1.00x)     | 3.8T     | 85.9       |
> | &nbsp;&nbsp; +SparseVLM    | 64      | 1068s (1.22x)          | 377s (1.31x)     | 1.3T     | 75.1       |
> | &nbsp;&nbsp; +Ours         | 64      | **685s (1.90x)**       | **235s (2.10x)** | **0.8T** | 86.2       |
> | LLaVA-NeXT-7B | 2880    | 2284s (1.00x)          | 1062s (1.00x)   | 12.6T    | 86.3       |
> | &nbsp;&nbsp; +SparseVLM    | 320     | 1872s (1.22x)          | 644s (1.65x)    | 2.5T     | 78.5       |
> | &nbsp;&nbsp; +Ours         | 320     | **921s (2.48x)**       | **360s (2.95x)**  | **1.2T** | 84.8       |
>
> We will include these extended efficiency results in the revised manuscript to highlight the advantages of VisionTrim better.
>
> ---

---

> ### Author Response · Authors · 2025-11-20
> **Response to Reviewer M569 (2/3)**
>
> ---
>
> > ### **`Weakness 2`: Lack of experiments exploring the detailed configuration.**
>
> We sincerely thank you for the valuable feedback. **Due to space constraints, the original configurations and ablation studies were placed in the Appendix of our initial submission.** Following your suggestion, we have summarized and conducted the key ablation experiments and hyperparameter settings as follows. Additionally, we will **release all code and hyperparameters** to support the practical application and reproducibility of VisionTrim in accelerating MLLMs.
>
> 1. > **`Weakness 2.1`: What percentage of vision tokens is pruned in the vision encoder and in the LLM decoder, respectively?**
>
>    In our default settings and the experiments presented in the main paper, we first set the token retention ratio $\alpha_1$ in the vision encoder to 50% and then dynamically adjust the token retention ratio $\alpha_2$ in the LLM decoder to achieve the target overall token reduction ratio, as shown in `Table 2`.
>
>    **Table 2. Detailed hyperparameter configurations used in VisionTrim.**
>
>    | Parameter                                          | Value                  |
>    | -------------------------------------------------- | ---------------------- |
>    | token retention ratio $\alpha_1$ in vision encoder | 50%                    |
>    | token retention ratio $\alpha_2$ in LLM decoder    | dynamically adjusted   |
>    | Inserted layer in ViT (pre-LLM)                    | 23rd (total 24 layers) |
>    | Inserted layer in LLM (within-LLM)                 | 2nd (total 32 layers)  |
>
> 2. > **`Weakness 2.2`: At what stages are they pruned?**
>
>    In our default settings and the experiments presented in the main paper, the proposed modules are inserted only between the penultimate and last layers of the vision encoder, and only between the second and third layers of the LLM, as shown in `Table 2`.
>
>    To further clarify the implementation details, we conducted additional ablation experiments with different LLM layer selections, as summarized in `Table 3`. Here, $k$ denotes the inserted LLM layer, and $\alpha_2$ represents the token retention rate after that layer. To maintain an overall token retention ratio of 75.0% throughout the LLM decoding stage (corresponding to 37.5% token retention over the full MLLM pipeline), we adaptively adjust $\alpha_2$ based on the inserted layer $k$. Our observations indicate that inserting the proposed modules into shallower layers generally yields better performance.
>
>    **Table 3: Ablation experiments of different inserted LLM layer $k$ across LLaVA-1.5-7B ($\alpha_1$ is set to 50%).**
>
>    | Settings                     | GQA ↑    | MMB ↑    | POPE ↑   |
>    | ---------------------------- | -------- | -------- | -------- |
>    | $k$ = 2, $\alpha_2$ = 73.3%  | 60.4     | **64.2** | **86.4** |
>    | $k$ = 8, $\alpha_2$ = 66.7%  | 60.2     | 64.0     | 86.0     |
>    | $k$ = 12, $\alpha_2$ = 60.0% | **60.5** | 63.8     | 85.5     |
>    | $k$ = 16, $\alpha_2$ = 50.0% | 58.4     | 62.9     | 84.4     |
>    | $k$ = 20, $\alpha_2$ = 33.3% | 57.3     | 63.3     | 84.1     |
>    | $k$ = 24, $\alpha_2$ = 0.0%  | 56.5     | 60.5     | 83.2     |
>
> 3. > **`Weakness 2.3`: What are the performance-efficiency trade-offs of pruning in the vision encoder *vs* in the LLM decoder?**
>
>    Thank you for your valuable advice. We conducted an ablation study on the performance–efficiency trade-offs of different token retention rates in the vision encoder ($\alpha_1$) versus the LLM decoder ($\alpha_2$) in LLaVA-1.5-7B, as shown in `Table 4`. We further observe that **aggressively pruning tokens in the vision encoder** can further reduce the model’s inference latency and KV cache usage, but at the cost of degraded performance. In contrast, **pruning more tokens in the LLM decoder** tends to preserve model performance, but results in increased inference latency and KV cache.
>
> **Table 4: Ablation study of performance–efficiency trade-offs for different token retention rates in the vision encoder ($\alpha_1$) versus the LLM decoder ($\alpha_2$) in LLaVA-1.5-7B (under an 88.9% overall token reduction ratio).**
>
> | $\alpha_1$ | $\alpha_2$ | POPE ↑   | End-to-End Inference Time ↓ | Prefill Time ↓ | KV Cache ↓ |
> | ---------- | ---------- | -------- | --------------------------- | -------------- | ---------- |
> | 88.8%      | 12.5%      | 86.0     | 806s                        | 284s           | 22.5%      |
> | 66.6%      | 16.7%      | 84.8     | 712s                        | 251s           | 18.8%      |
> | 50.0%      | 22.2%      | **86.2** | 685s                        | 235s           | 17.2%      |
> | 16.7%      | 66.6%      | 83.4     | 657s                        | 224s           | 16.8%      |
> | 12.5%      | 88.8%      | 82.8     | 643s                        | 218s           | 16.2%      |
>
> We will include the above experimental results in the main paper to better highlight the advantages of VisionTrim.
>
> ---

---

> ### Author Response · Authors · 2025-11-20
> **Response to Reviewer M569 (3/3)**
>
> ---
>
> > ### **`Weakness 3`: Minor presentation issues: the intuition behind some equations, especially those related to local spatial continuity, could be further elaborated.**
>
> We sincerely thank you for the valuable suggestion. Existing similarity-based methods typically measure token redundancy by considering only the feature similarity between tokens. Intuitively, a token may be deemed highly redundant if it is very similar to another visual token. However, tokens are characterized not only by their embeddings but also by their spatial positions. **For example, even if a token is similar to another token located far away, it may not be truly redundant due to their large positional distance.** Ignoring this aspect can lead to the blurring or loss of spatial information.
>
> Therefore, in designing **local spatial continuity**, we consider not only the feature similarity between tokens $\kappa\_{\text{feat}}^{xy,uv}$ but also their spatial proximity $\kappa_{\text{pos}}^{xy,uv}$, defined as follows:
> $$
> \kappa_{feat}^{xy,uv} = -\left(\frac{\|F_{xy} - F_{uv}\|}{w_1\sigma_f}\right)^2, \kappa_{pos}^{xy,uv} = -\left(\frac{\|P_{xy} - P_{uv}\|}{w_2\sigma_p}\right)^2.
> $$
> Thus, by computing a weighted combination of the two, we obtain the composite metric ${\kappa^{*}}^{xy,uv}$, which is then used to select the tokens that preserve the maximum visual information:
>
> $$
> {{\kappa}^*}^{xy,uv} = \kappa_{feat}^{xy,hw} + w_3 \kappa_{pos}^{xy,hw}.
> $$
>
> ---
>
> > ### **`Question 1`: Clarification of the used [fire] symbol in Figure 3(a) and the [snowflake] symbol in both Figure 3(a) and Figure 1.**
>
> We sincerely appreciate you for pointing out this presentation issue. Both the proposed DVTS and TGVC modules function as training-free, plug-and-play components. **There should not be a [fire] symbol over TGVC in Figure 3(a).** The [snowflake] symbol indicates that a module is frozen. Since VisionTrim does not require fine-tuning of the overall MLLM pipeline, the [snowflake] symbols over other modules in both Figure 1 and Figure 3(a), such as the vision encoder and projector, are correct. We have fixed this issue in the revised version. We sincerely apologize for the confusion caused by our oversight and greatly appreciate your careful review.
>
> ---
>
> We believe that the revisions incorporated—an expanded analysis of the overall end-to-end latency, additional ablation experiments on module configurations, and more detailed elaborations of several equations—will substantially strengthen the paper. We hope that our responses have adequately addressed your concerns and sincerely appreciate your constructive feedback once again.

---

> > ### Comment · Reviewer_M569 · 2025-11-27
> > **Response to authors**
> >
> > I appreciate the authors' response to my concerns and decided to increase my score to 6.

---

> ### Author Response · Authors · 2025-11-26
> **Look forward to your post-rebuttal feedback**
>
> Dear Reviewer M569,
>
> Thank you again for your valuable time and effort in reviewing our submission. We hope that our responses have sufficiently addressed the concerns and points you raised, and that the corresponding clarification and improvements may support a positive re-evaluation of our work, for which we would be deeply grateful.
>
> As the discussion stage draws to a close, we would like to respectfully follow up and would greatly appreciate any further feedback, concerns, or suggestions you may have, so that we can address them accordingly.
>
> We sincerely appreciate your continued engagement and contributions throughout the review and discussion process.
>
> Best regards,
>
> The Authors

---

> ### Author Response · Authors · 2025-11-27
> **Thank you for your encouraging feedback!**
>
> Dear Reviewer M569,
>
> We sincerely appreciate your positive feedback and are truly delighted that our responses have addressed your concerns. Your support and positive recommendation mean a great deal to us, and your constructive comments have been immensely valuable in helping us improve the paper. Thank you once again for your time and thoughtful review!
>
> Best regards,
>
> The Authors

---

### Official Review · Reviewer_r8MU · 2025-10-28

**Soundness:** 2
**Presentation:** 2
**Contribution:** 3
**Rating:** 6
**Confidence:** 4

**Summary:**

The paper proposed to prune the visual tokens to accelerate MLLM computation, referred as VisionTrim, including two training-free plug-and-play modules:

1)	Donimant Vision Token Selection (DVTS), which defines a global importance score that is the average attention scores w.r.t. the [CLS] token, and a local importance score that is the average affinity score within its neighboring tokens. These two importance scores are averaged by their variance to select the top K informative tokens.

2)	Text-Guided Vision Complement (TGVC), which assumes an associated textual descriptions/prompts related to the image and uses the CLIP score between the visual and text tokens to select top-R tokens as clustering centers.

These modules can be plugged in to the vision encoding stage or the LLM decoding stage. Experiments on LLaVA-NeXT-7B, Video-LLaVA-7B, and Qwen2.5-VL-7B models show that the proposed VisionTrim achieve competitive performance on multiple tasks after pruning 78% to 94% visual tokens.

**Strengths:**

The proposed DVTS and DVTS models are easy to implement and reproduce, which empirically show competitive performance even after pruning a large portion of visual tokens. The experiments are thorough and convincing. This may be inspiring to practical MLLM systems.

**Weaknesses:**

What if these is no associated textual prompts available? Or the textual clues are noisy and misleading?

Given the same image, if the textual prompts are varying in different tasks, the image has to be processed per textual prompt since the TGVC results may be quite different, right?

**Questions:**

How to choose top-K in DVTS  and top-R in TGVC?

---

> ### Author Response · Authors · 2025-11-20
> **Response to Reviewer r8MU (1/2)**
>
> We sincerely thank Reviewer r8MU for the careful reading, for recognizing the ease of implementation and reproducibility of DVTS and TGVC modules, the superior performance of VisionTrim under ultra-low-token scenarios, and the overall comprehensiveness of experiments. Below, we address each concern and outline the concrete revisions we have made.
>
> ---
>
> > ### **`Weakness 1`: What if there are no associated textual prompts available? Or are the textual clues noisy and misleading?**
>
> Thank you very much for your insightful feedback regarding the robustness of VisionTrim when textual prompts are unavailable, irrelevant, or misleading. In the VisionTrim framework, token compression is carried out **in two sequential stages:** (1) dominant visual tokens are selected via a dual-attention mechanism integrating global semantics and local spatial continuity; (2) among the discarded tokens, text-guided complementation is applied to restore essential text-relevant cues when available.
>
> **1. Scenario A: No Associated Textual Prompts.** In the absence of textual prompts, VisionTrim can operate **in a text-agnostic manner** by relying solely on the DVTS module. DVTS selects visual tokens using the average attention from the [CLS] token and local affinity scores among neighboring tokens. As shown in `Table 1`, our ablation study demonstrates that the DVTS-only configuration preserves most of the model’s accuracy while substantially improving inference efficiency. This confirms that VisionTrim remains robust even when textual signals are absent.
>
> **Table 1. Additional performance-efficiency experiments on different components in LLaVA-1.5-7B (under an 88.9% token reduction ratio).**
>
> | Method             | #Tokens | POPE ↑ | Total Inference Time ↓ | Prefill Time ↓ |
> | ------------------ | ------- | ------ | ---------------------- | -------------- |
> | Vanilla            | 576     | 85.9   | 1303s (1.00x)          | 494s (1.00x)   |
> | *w/* DVTS (only)   | 64      | 84.6   | 670s (1.94x)           | 227s (2.18x)   |
> | DVTS + TGVC (full) | 64      | 86.2   | 685s (1.90x)           | 235s (2.10x)   |
>
> **2. Scenario B: Textual Prompts are Irrelevant or Misleading.** Even when the full pipeline (DVTS + TGVC) is used with poor-quality prompts, VisionTrim remains robust due to the following mechanisms:
>
> - **When prompts are unrelated to the image,** the text-image similarity distribution tends to be uniform or random. Consequently, the initialization of cluster centers statistically behaves like **random sampling** (akin to K-Means initialization). Owing to the high semantic redundancy of visual tokens, this naturally leads VisionTrim to behave like an **unsupervised visual clustering** method—an approach shown to be effective in prior token-merging works such as ToMe [1]. Since clustering assignments are driven strictly by visual similarity, object-related tokens naturally group together, preserving semantic integrity even in the absence of meaningful textual cues.
> - **When textual clues are noisy or misleading,** the text–image similarity still reflects basic feature-space alignment. For example, if the prompt is "car" but the image contains none, cosine similarities across all patches remain uniformly low. Since the algorithm cannot fabricate absent features, the selection mechanism naturally falls back to tokens with the highest relative projection in the feature space—typically salient objects or complex textures with high feature norms. **Thus, misleading text rarely diverts attention completely from dominant visual content.** **Furthermore, VisionTrim avoids over-reliance on textual validity:** in TGVC, text is used only for initialization, while the subsequent clustering assignment is governed by intrinsic visual similarity. **Since VisionTrim performs token merging rather than directly pruning**, even imperfect initialization causes only minor spatial shifts without discarding essential semantics. Consequently, misleading textual prompts do not lead to semantic information loss, and the model can robustly recover the dominant visual features.
>
> Additionally, we highlight **the hierarchical nature of VisionTrim’s compression pipeline**: dominant visual content is preserved first (DVTS), and textual-related information is incorporated afterward (TGVC). Since the key visual information is already secured in the first stage, the potential noise introduced in the text-guided stage has a minimal influence on overall performance.
>
> ---
>
> *[1] Bolya D, Fu C Y, Dai X, et al. Token merging: Your vit but faster[J]. arXiv preprint arXiv:2210.09461 (2022).*

---

> ### Author Response · Authors · 2025-11-20
> **Response to Reviewer r8MU (2/2)**
>
> ---
>
> > ### **`Weakness 2`: Given the same image, if the textual prompts vary in different tasks, the image has to be processed per textual prompt since the TGVC results may be quite different.**
>
> Thank you for your valuable feedback. If the textual prompts differ across tasks while the image remains the same, then indeed the image needs to be processed per prompt in the TGVC module. However, since the dominant visual tokens have already been selected in the DVTS stage, the number of remaining tokens passed to the TGVC stage has already been reduced to some extent.  **Moreover, the DVTS processing is identical for the same image regardless of the textual prompt.**
>
> Therefore, for different prompts, we only need to perform the TGVC operation on **this reduced subset of image tokens**, which effectively avoids redundant computation and requires a small amount of additional visual–textual similarity computation. We measured that the token merging process in TGVC takes less than `1 ms` per sample on the POPE benchmark, introducing negligible overhead.
>
> ---
>
> > ### **`Question 1`: How to choose top-K in DVTS and top-R in TGVC?**
>
> To facilitate the practical application and reproducibility of VisionTrim for accelerating MLLMs, we provide detailed hyperparameter configurations. In general, since the core design philosophy of VisionTrim is to preserve the primary visual information in the DVTS stage and then perform complementary refinement in the TGVC stage, we allocate a larger portion of tokens to DVTS. Typically, we set the ratio $K$:$R$ = 3:1, as shown in `Table 2`.
>
> **Table 2. Default token configurations for $K$ in the DVTS module and $R$ in the TGVC module.**
>
> | DVTS  : TGVC | Retained 32 tokens | Retained 64 tokens | Retained 128 tokens | Retained 192 tokens |
> | ------------ | ------------------ | ------------------ | ------------------- | ------------------- |
> | $K$ : $R$    | 24 : 8             | 48 : 16            | 96 : 32             | 144 : 48            |
>
> We further conduct an ablation study on different ratios between $K$ in the DVTS module and $R$ in the TGVC module, as shown in `Table 3`. We observe that **if $K$ in DVTS is too small**, the model performance drops noticeably because insufficient dominant visual information is preserved. **Conversely, if $R$ in TGVC is too small**, the pruning results from DVTS cannot be effectively complemented, causing some critical text-related information to be lost and leading to a performance decline.
>
> In contrast, a more balanced and progressive two-stage pruning strategy ($K$:$R$ = 3:1) achieves the best performance. These results not only validate the effectiveness of our token-count configuration, but also demonstrate the advantage of **jointly leveraging both DVTS and TGVC** to improve efficiency while preserving accuracy.
>
> **Table 3. Ablation study on different ratios between $K$ in the DVTS module and $R$ in the TGVC module for LLaVA-1.5-7B (under an 88.9% overall token reduction ratio, retained 64 tokens).**
>
> | $K$ in DVTS | $R$ in TGVC | TextVQA ↑ | MMB ↑    | POPE ↑   |
> | ----------- | ----------- | --------- | -------- | -------- |
> | 8           | 56          | 51.5      | 57.4     | 81.2     |
> | 16          | 48          | 53.4      | 59.8     | 83.8     |
> | 32          | 32          | 54.3      | 60.2     | 84.4     |
> | 48          | 16          | **56.8**  | **63.0** | **86.2** |
> | 56          | 8           | 55.5      | 60.3     | 85.0     |
>
> ---
>
> We believe that the revisions outlined—the analysis and handling of cases where textual prompts are absent, noisy, or misleading, as well as the more detailed hyperparameter configurations for DVTS and TGVC—will substantially strengthen the paper. We hope that our responses have adequately addressed your concerns and sincerely thank you once again for your insightful comments.

---

> > ### Comment · Reviewer_r8MU · 2025-11-22
> >
> > Thank you for the detailed response. I agree that the revision on the usage of textual prompts will clarify the concerns the readers may have.

---

> ### Author Response · Authors · 2025-11-24
> **Thank you for your positive feedback!**
>
> Dear Reviewer r8MU,
>
> We sincerely appreciate your positive feedback and are delighted that our responses have addressed your concerns. In the revised manuscript, we have added a subsection on the usage of textual prompts, highlighted in `blue`, in `Section 4.3` on page 9. Your support and inclination toward acceptance are truly encouraging, and we deeply value your thoughtful review and constructive comments.
>
> Best regards,
>
> The Authors

---

### Official Review · Reviewer_RMDZ · 2025-11-01

**Soundness:** 3
**Presentation:** 3
**Contribution:** 3
**Rating:** 6
**Confidence:** 4

**Summary:**

The manuscript identifies two drawbacks of existing token reduction methods:

(i) they focus on only one of visual encoding or LLM decoding processes,

(ii) they neglect the role of text queries during the token pruning process.

The authors propose VisionTrim to overcome these drawbacks, where tokens are selected based on both visual and textual information (using DVTS and TGVC respctively), and the proposed modules are applied both after visual encoding and during LLM decoding.

The proposed solutions for both drawbacks are validated in the experiment section. Both text-guided token selection and multi-stage application of token selection strategy are shown to be effective in improving the performance of low-token scenario. Finally, compared with existing approaches, VisionTrim demonstrates competitive performance.

**Strengths:**

1. The proposed method is well-motivated. Two key drawbacks of existing token pruning methods are identified, respectively the neglection of the role of text in visual token selection, and the application of token pruning on only one stage of visual encoding or language decoding. This provides a good review of existing methods and provides guidance to future research on token pruning for training-free mllm acceleration.

2. The component designs are reasonable and thoroughly ablated in Table 4 and 5, including the local affinity in visual token selection, text-guided selection, and multi-stage application of token pruning, showing sound effectiveness of these components.

3. The performance of VisionTrim is competitive against existing token pruning methods.

**Weaknesses:**

1. Despite the effectiveness of the proposed strategies resolving the two drawbacks of token pruning methods, in Table 5, it seems the ensemble strategy is playing a vital role in the strong performance. This is intriguing and I believe it deserves more explanation, both in terms of how the adaptive weighting mechanism is designed, as well as why it is so effective.

2. Different components added to the final model may have different latencies, which the manuscript fails to provide.

3. The usage of the first generated token as the natural measure of global semantic significance over all image tokens in LLM decoding stage does not make sense to me. Why is this strategy chosen?

**Questions:**

1. I am not sure about the term CUDA time, is it the same as latency, in terms of the time spent on processing the samples?

---

> ### Author Response · Authors · 2025-11-20
> **Response to Reviewer RMDZ (1/2)**
>
> We sincerely thank Reviewer RMDZ for the careful reading, for recognizing the clear motivation of our method, the effectiveness of our component designs, and the competitive performance of VisionTrim. Below, we respond to each concern and detail the concrete revisions.
>
> ---
>
> > ### **`Weakness 1`: How the adaptive weighting mechanism is designed and why it is so effective.**
>
> Thank you for your valuable suggestion. We address and clarify this concern from the following two perspectives.
>
> 1. > **`Weakness 1.1`: How is the adaptive weighting mechanism designed?**
>
>    Regarding how to combine the global and local importance scores, the proposed adaptive weighting mechanism is based on the observation that their variances reflect confidence: **a smaller variance indicates a more concentrated and reliable distribution, whereas a larger variance suggests a more dispersed distribution with higher uncertainty.** We compute the variances of the global importance scores $\\{S_i^g\\}\_{i=1}^{N}$ and the local importance scores $\\{S_i^l\\}_{i=1}^{N}$ as $\sigma_g^2$ and $\sigma_l^2$, respectively, and assign weights using the following formula:
>        $$
>        S_i = \alpha {S}^g_i + (1-\alpha) S^l_i, \ \ \text{where} \ \ \alpha = \sigma_l^2 \mathord{\left/ \vphantom{\sigma_l^2 \sigma_g^2} \right.} (\sigma_g^2+\sigma_l^2).
>        $$
>    **When the distribution of an indicator is more dispersed (i.e., has a higher variance), it can have an outsized influence on the weighted result.** In such cases, its weight should be reduced to prevent a single indicator from dominating, thereby ensuring that both types of indicators are considered in a balanced manner. This adaptive approach eliminates the need to manually set the alpha parameter, as the weights are automatically determined based on the data characteristics.
>
> 2. > **`Weakness 1.2`: Why does the adaptive weighting mechanism demonstrate such high effectiveness?**
>
>    Adaptive weighting generalizes well across different images, layers, and tasks. Since feature distributions vary greatly—simple images may rely more on global attention while complex ones require stronger local cues—the adaptive method automatically adjusts weights based on reliability. When global attention is more trustworthy, its weight increases; when local importance is more reliable, it receives greater emphasis. **This dynamic adjustment effectively leverages complementary information while reducing the impact of noisy indicators, improving robustness.**
>
>    In contrast, fixed weighting cannot adapt to sample- or layer-level differences, the geometric mean is overly sensitive to low values, and the element-wise maximum may ignore one indicator, risking noise amplification or information loss. As shown in `Table 1`, combining global semantics and local spatial continuity with adaptive weighting outperforms other ensemble strategies across multiple benchmarks.
>
> **Table 1. More ablation study of various ensemble strategies in the DVTS module of LLaVA-1.5-7B (under an 88.9% overall token reduction ratio).**
>
> | Ensemble Strategy    | GQA ↑    | MMB ↑    | SQA ↑    | MME ↑    | POPE ↑   |
> |-|-|-|-|-|-|
> | Element-wise Maximum | 53.4     | 58.7     | 70.0     | 1702     | 77.6     |
> | Fixed Weighting      | 56.0     | 57.4     | 68.2     | 1695     | 77.0     |
> | Geometric Mean       | 55.2     | 56.5     | **71.1** | 1631     | 80.2     |
> | Adaptive Weighting   | **58.8** | **63.0** | 71.0     | **1780** | **86.2** |
>
> ---
>
> > ### **`Weakness 2`: Different components added to the final model may have different latencies.**
>
> Thank you for your valuable suggestion. **We conducted additional performance–efficiency experiments on different components added to the final model, as shown in `Table 2`.** We observe that using only the DVTS module further improves inference efficiency, but leads to a larger performance drop because text-related information is not considered. In contrast, using only the TGVC module preserves performance more effectively, but yields a smaller improvement in inference efficiency due to the increased token-level similarity computations involved in clustering. **By combining both modules, the final model achieves a better trade-off between performance and efficiency.**
>
> **Table 2. Extended performance-efficiency analysis of different components in LLaVA-1.5-7B (under an 88.9% overall token reduction ratio).**
>
> |Method|#Tokens|POPE ↑|Total Inference Time ↓|Prefill Time ↓
> |-|-|-|-|-|
> | Vanilla|576|85.9| 1303s (1.00x)| 494s (1.00x)|
> | *w/* DVTS (only)   | 64 | 84.6   | 670s (1.94x)| 227s (2.18x)   |
> | *w/* TGVC (only)   | 64| 85.0   | 741s (1.76x)| 263s (1.88x)   |
> | DVTS + TGVC (full) | 64| 86.2   | 685s (1.90x)| 235s (2.10x)   |
>
> We have included these extended efficiency results and analyses, highlighted in `blue`, in `Appendix D.3.8` and in `Table 10` on page 21 of the revised manuscript.
>
> ---

---

> ### Author Response · Authors · 2025-11-20
> **Response to Reviewer RMDZ (2/2)**
>
> ---
>
> > ### **`Weakness 3`: Why chooses the first generated token as the natural measure of global semantic significance over all image tokens in LLM decoding stage?**
>
> Thank you for your valuable feedback. First, during LLM decoding, generated tokens attend to all image tokens via the attention mechanism, **highlighting the regions that contribute most to the output.** This attention distribution serves as an effective criterion for token selection. Unlike indicators based solely on visual encoder features, it integrates both visual and linguistic semantics, aligning more closely with the task objective and enabling more accurate identification of image regions that influence the final output.
>
> Second, we use the first generated token for token compression **to reduce subsequent LLM decoding costs as early as possible.** By pruning immediately after the first token is generated, all remaining decoding steps operate on the compressed sequence, significantly lowering computational complexity. While waiting for additional tokens could provide richer attention information, it would delay compression, forcing the model to process the full sequence for further decoding and reducing the speedup. Therefore, using the first token strikes an optimal balance between information quality and computational efficiency.
>
> ---
>
> > ### **`Question 1`: Clarification on the term "CUDA time".**
>
> We sincerely apologize for the confusion. The term "CUDA time" refers to the LLM decoding time during the forward propagation of the MLLM. To more comprehensively evaluate the impact of VisionTrim on reducing inference latency, we also conducted additional efficiency experiments measuring both the overall end-to-end inference time and the prefill time on a single NVIDIA A100 GPU, as shown in `Table 3`. VisionTrim achieves a **2.48×** speedup in overall end-to-end latency and a **2.95×** speedup in prefill time on LLaVA-NeXT-7B, while maintaining robust performance or even achieving improvements, substantially outperforming existing methods in both accuracy and efficiency. We will clarify the utilized efficiency metrics and include these extended efficiency results in the revised manuscript to more clearly highlight the advantages of VisionTrim.
>
> **Table 3. More efficiency analysis on the POPE benchmark (under an 88.9% token reduction ratio).**
>
> | Method        | #Tokens | Total Inference Time ↓ | Prefill Time ↓   | FLOPs ↓  | Accuracy ↑ |
> | ------------- | ------- | ---------------------- | ---------------- |-------- | ---------- |
> | LLaVA-1.5-7B  | 576     | 1303s (1.00x)          | 494s (1.00x)     | 3.8T     | 85.9       |
> | &nbsp;&nbsp; +SparseVLM    | 64      | 1068s (1.22x)          | 377s (1.31x)     | 1.3T     | 75.1       |
> | &nbsp;&nbsp; +Ours         | 64      | **685s (1.90x)**       | **235s (2.10x)** | **0.8T** | 86.2       |
> | LLaVA-NeXT-7B | 2880    | 2284s (1.00x)          | 1062s (1.00x)   | 12.6T    | 86.3       |
> | &nbsp;&nbsp; +SparseVLM    | 320     | 1872s (1.22x)          | 644s (1.65x)    | 2.5T     | 78.5       |
> | &nbsp;&nbsp; +Ours         | 320     | **921s (2.48x)**       | **360s (2.95x)**  | **1.2T** | 84.8       |
>
> ---
>
>  We believe that the revisions outlined— expanded analysis of the ensemble strategy, additional experiments evaluating the performance-efficiency trade-offs of different components, and more detailed clarifications of the presented terms—will significantly enhance the paper. We hope our responses have adequately addressed your concerns and sincerely thank you again for the constructive comments.

---

> ### Author Response · Authors · 2025-11-26
> **Look forward to your post-rebuttal feedback**
>
> Dear Reviewer RMDZ,
>
> Thank you again for your valuable time and effort in reviewing our submission. We hope that our responses have sufficiently addressed the concerns and points you raised. As the discussion stage draws to a close, we would like to respectfully follow up and would greatly appreciate any further feedback, concerns, or suggestions you may have, so that we can address them accordingly.
>
> We sincerely appreciate your continued engagement and contributions throughout the review and discussion process.
>
> Best regards,
>
> The Authors

---

### Author Response · Authors · 2025-11-29
**General Response**

**Dear Reviewers, ACs, and SACs,**

We sincerely thank all reviewers for their thoughtful comments and constructive feedback throughout the discussion period.

---

We are grateful for the reviewers' recognition of our work as **a unified training-free token compression framework** designed to accelerate MLLM inference. VisionTrim introduces **two plug-and-play modules (DVTS and TGVC)**, which can be applied to both the visual encoding and LLM decoding stages to better preserve essential visual information. VisionTrim **consistently outperforms existing token-pruning methods** across a wide range of compression ratios, benchmarks, and MLLM families, while **significantly reducing inference latency**. We believe that VisionTrim will advance practical MLLM deployment in real-world applications and inform future research on training-free token pruning.

Overall, we are encouraged by the reviewers' positive feedback, which highlights:

- **Well-motivated**: Provides a solid review of existing methods, and the insights—especially the idea of jointly tailoring the vision encoder and LLM decoder—are highly instructive for practical MLLM systems and industrial applications. (Reviewers `RMDZ`, `r8MU`, `8p5d`)
- **Reasonable design and thorough experiments**: VisionTrim introduces a unified framework applicable to the full MLLM pipeline, incorporating local affinity–based token selection, text-guided selection, and multi-stage pruning. Extensive testing and ablations across various MLLM families further validate its effectiveness. (Reviewers `r8MU`, `M569`, `8p5d`)
- **Superior performance**: VisionTrim demonstrates strong results compared with existing token-pruning methods across diverse benchmarks, particularly under ultra-low vision token settings. (Reviewers `RMDZ`, `r8MU`, `M569`)
- **Well-written**: Figures, tables, and qualitative results are clear, intuitive, and effectively illustrate the proposed methods. (Reviewers `M569`, `8p5d`)

---

To address the reviewers' concerns, we have conducted several additional experiments and analyses, including:

- **Provide a more thorough efficiency analysis** that evaluates both overall end-to-end latency and prefill time. (Reviewers `RMDZ`, `M569`)
- **Expand the detailed configurations**, including the pruning ratios for the vision encoder and LLM decoder, the pruning stages, and the associated performance–efficiency trade-offs. (Reviewers `M569`, `8p5d`)
- **Offer additional explanation of the intuition** behind the design of the adaptive weighting mechanism and the local spatial continuity. (Reviewers `RMDZ`, `M569`)
- **Clarify the usage of textual cues** in the text-guided vision complement and during the LLM decoding stage. (Reviewers `RMDZ`, `r8MU`)

---

**Summary of Revisions:**

- Provided a more comprehensive efficiency analysis in `Section 4.4` and `Table 9`.
- Incorporated the performance-efficiency trade-offs on different components in `Appendix D.3.8` and `Table 10`.
- Included a subsection on individual case performance analysis in `Appendix A` and `Figure 7`.
- Added a subsection on the usage of textual prompts in `Section 4.3`.

These updates are highlighted in `blue` for easy reference in the revised manuscript. We deeply appreciate the insightful and valuable comments provided by all reviewers, which have greatly helped improve our paper.

---

Thank you again to all reviewers and the AC for your time, effort, and constructive suggestions!

Best regards,

The Authors

---

### Author Response · Authors · 2025-12-01
**Summary of Author–Reviewer Discussion for the Area Chair**

Dear Area Chair,

We would like to briefly summarize the productive author-reviewer discussions during the rebuttal period.

---

In the initial reviews, three reviewers (`RMDZ`, `r8MU`, `8p5d`) gave positive scores (6), while one reviewer (`M569`) assigned a borderline score (4).

After extensive and constructive discussions with the reviewers, the overall ratings progressed from **(6, 6, 6, 4)** to **(6, 6, 6, 6)**:

- **Reviewer `M569`** acknowledged that our response addressed the concerns and explicitly **raised the score from 4 → 6** *prior to the public identification leak caused by the OpenReview API issue* (reported at 10:09 AM EST on Nov 27 by ICLR 2026 Workflow Chair), as evidenced by the response timeline:

  > “I appreciate the authors’ response to my concerns and decided to **increase my score to 6**.”
  >  *Timestamp:* 03:40 AM EST, Nov 27.

The remaining reviewers maintained their positive recommendations, noting that their concerns had been effectively addressed and expressing agreement with our responses and revisions—*also prior to the public identification leak caused by the OpenReview API issue*—as summarized below.

- **Reviewer `r8MU`** kept **the positive score (6)** and acknowledged the adequacy of our revision regarding the usage of textual prompts:

  > “Thank you for the detailed response. I agree that the revision on the usage of textual prompts will clarify the concerns the readers may have.”
  >  *Timestamp:* 10:03 AM EST, Nov 22.

- **Reviewer `8p5d`** also maintained **the positive score (6)**, affirming that our rebuttal effectively resolved the concerns:

  > “Thank you for the comprehensive rebuttal. The clarifications provided effectively resolve my concerns. … As my initial overall assessment was positive, I will maintain my score.”
  >  *Timestamp:* 23:08 PM EST, Nov 20.

- **Reviewer `RMDZ`**, in the initial review, highlighted that *“The proposed method is well-motivated… The component designs are reasonable and thoroughly ablated… The performance of VisionTrim is competitive...”* and consistently maintained **the positive score (6)** throughout the rebuttal period. In response, we expanded the analysis of the ensemble strategy, conducted thorough experiments on the trade-offs of different components, and clarified the intuition behind VisionTrim and the presented terms, which we believe adequately address the primary concern.

---

We regret the impact of the OpenReview bug and sincerely appreciate your time and effort.

Best regards,

The Authors

---

### Meta-Review · Area_Chair_C4h8 · 2026-01-06

**Summary:**

Overall, three reviewers out of four are marginally positive before rebuttal, and one marginal negative. The concerns are relatively minor, specifically linked to details linked to latency definition, design choices, missing details and readability.

**Reviewer Concerns:**

Most of the concerns are addressed; one in the AC opinion, still remain :
- "The usage of the first generated token as the natural measure of global semantic significance over all image tokens in LLM decoding stage does not make sense to me" by RMDZ. Although the authors reply that the choice is for efficiency purposes, this is not a valid reason for guaranteeing it is the best option, but rather an arbitrary choice, that works in the tested scenarios. As such, it remains an unclear point, with little grounding.

Besides, on the AC perspective, the evaluation and comparison could have been richer: for example, works like Wang et al. (2024b) are referenced but not compared to.

**Reviewer Scores:**

The final assessment would realistically be aligned to a marginal accept by all the reviewers.

---

### Decision · Program_Chairs · 2026-01-26

Accept (Poster)